# Toward Enhancing Representation Learning in Federated Multi-Task Settings

**Mehdi Setayesh, Mahdi Beitollahi, Yasser H. Khalil, and Hongliang Li**
Huawei Noah's Ark Lab, Montreal, Canada
`mehdi.setayesh1@huawei.com`  `mahdi.beitollahi@h-partners.com`
`yasser.khalil1@huawei.com`   `hongliang.li2@huawei.com`

## Abstract

Federated multi-task learning (FMTL) seeks to collaboratively train customized models for users with different tasks while preserving data privacy. Most existing approaches assume model congruity (i.e., the use of fully or partially homogeneous models) across users, which limits their applicability in realistic settings. To overcome this limitation, we aim to learn a shared representation space across tasks rather than shared model parameters. To this end, we propose *Muscle loss*, a novel contrastive learning objective that simultaneously aligns representations from all participating models. Unlike existing multi-view or multi-model contrastive methods, which typically align models pairwise, Muscle loss can effectively capture dependencies across tasks because its minimization is equivalent to the maximization of mutual information among all the models' representations. Building on this principle, we develop *FedMuscle*, a practical and communication-efficient FMTL algorithm that naturally handles both model and task heterogeneity. Experiments on diverse image and language tasks demonstrate that FedMuscle consistently outperforms state-of-the-art baselines, delivering substantial improvements and robust performance across heterogeneous settings.

## 1 Introduction

Federated learning (FL) enables a group of users to collaboratively train models while preserving privacy by not sharing their local data (Konečný et al., 2016; AbdulRahman et al., 2021). Most conventional FL algorithms assume that users share the same model architecture (Smith et al., 2017; Li et al., 2019; T Dinh et al., 2020; Li et al., 2021b) or perform the same task (Diao et al., 2021; Alam et al., 2022; Zhu et al., 2022; Setayesh et al., 2023), which limits their applicability in diverse real-world settings (Cai et al., 2024; Wang et al., 2024). As illustrated in Figure 1, practical FL scenarios may involve users with heterogeneous model architectures and tasks, each using a task-specific local dataset. For example, with recent advances in foundation models (FMs), users can select a pre-trained FM based on their resource constraints and task requirements, and then fine-tune it using a local dataset (Bommasani et al., 2022; Shao et al., 2024; Zheng et al., 2025). Task heterogeneity in FL scenarios has motivated the development of various federated multi-task learning (FMTL) algorithms (Park et al., 2021; He et al., 2024; Chen et al., 2023; Lu et al., 2024; Jia et al., 2024).

Early works in FMTL focused on personalization tasks (i.e., non-IID data across users) and aimed to learn a customized model with the same architecture for each user (Smith et al., 2017; Marfoq et al., 2021). Recent studies have addressed a wider range of tasks by dividing users' models into a shared encoder and task-specific predictors (Jia et al., 2024; Lu et al., 2024). As a result, most existing FMTL algorithms assume that users employ fully or partially homogeneous model architectures. From a broader perspective, our central idea is that sharing model parameters, whether fully or partially, ultimately aims to establish a shared representation space for effective knowledge transfer among users' tasks. Therefore, the objective of an FMTL algorithm can be reframed as learning a shared representation space across tasks.

Contrastive learning (CL) is a widely used technique for learning a shared representation space (He et al., 2020; Misra & Maaten, 2020; Chen et al., 2020). The core idea of CL is to bring similar instances, known as positives, closer together in the representation space while pushing dissimilar ones, known as negatives, farther apart (Radford et al., 2021). One popular CL loss function is InfoNCE (Oord et al., 2019), which aligns the representations of two models. For scenarios with more

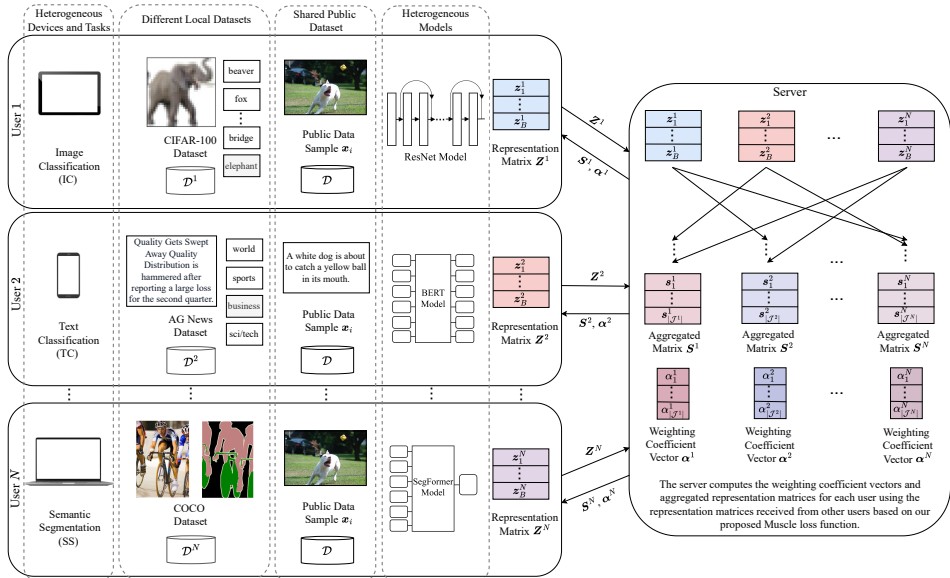

Figure 1: FedMuscle overview. Users may have heterogeneous models, tasks, and local datasets. A shared public dataset is used by all users to align the representation spaces of their models via the Muscle loss function.

than two models, prior works typically adopt pairwise alignment approaches, applying the InfoNCE loss to each pair of models (Yang et al., 2022; Girdhar et al., 2023; Xue et al., 2024a). However, pairwise alignment cannot effectively capture the dependencies among representations obtained from all models (Koromilas et al., 2025). To address this limitation, Cicchetti et al. (2025) recently proposed a Gramian-based contrastive loss that aligns all models' representations simultaneously. Nevertheless, their approach lacks theoretical justification.

In this paper, we propose *Muscle* (**Mu**lti-task/modal **S**ystematic **C**ontrastive **Le**arning), a novel CL loss function that effectively captures dependencies among the representations of all models in multi-model scenarios. This loss incorporates theoretically grounded weighting coefficients to emphasize more dissimilar negatives, an important mechanism absent in prior works. We show that minimizing the proposed Muscle loss is equivalent to maximizing a lower bound on the mutual information (MI) between the models' outputs, thereby enhancing knowledge transfer across tasks. Furthermore, we design a novel FMTL algorithm called *FedMuscle* by leveraging the Muscle loss. As shown in Figure 1, in FedMuscle, users transmit their private knowledge on a shared public dataset to the server rather than sharing their local datasets or model parameters. Not revealing local model parameters provides an additional layer of privacy protection, especially when users employ pre-trained FMs (Du et al., 2025). Upon receiving low-dimensional representations of public data from users, the server computes the weighting coefficients and aggregated representations that each user needs to align its representation space with those of others by minimizing the Muscle loss.

In summary, (1) we propose Muscle loss, a CL objective that applies to representations from any number of models and systematically captures dependencies among them. (2) We develop FedMuscle, a novel FMTL algorithm that addresses both model and task heterogeneity across users by aligning their representation spaces using Muscle loss. (3) We empirically validate the effectiveness of FedMuscle on various computer vision (CV) and natural language processing (NLP) tasks, demonstrating its superior performance over state-of-the-art baselines under settings with model and task heterogeneity. (4) We show that our proposed systematic CL framework can be seamlessly integrated into multi-modal FL algorithms, such as CreamFL (Yu et al., 2023), to enhance performance by replacing the heuristic knowledge distillation approach with Muscle loss.

## 2 RELATED WORK

**Federated Multi-Task Learning**   Various FMTL algorithms have been proposed to address task heterogeneity in FL. For example, FeSTA (Park et al., 2021), FedBone (Chen et al., 2023), FedHCA$^2$ (Lu et al., 2024), FedLPS (Jia et al., 2024), and FedDEA (Wei et al., 2025) have been introduced to enhance the generalization performance of all tasks by leveraging knowledge from re-

lated tasks. However, all of these FMTL algorithms assume model congruity, meaning that all users employ fully or partially homogeneous model architectures. Another line of research is federated semi-supervised learning (FSSL) (Jeong et al., 2021), with algorithms such as CoFED (Cao et al., 2023) and SAGE (Liu et al., 2025), where users collaboratively generate pseudo-labels for a public dataset. However, most FSSL approaches can only handle personalization tasks.

**Model Heterogeneity in FL**   Model heterogeneity in FL may arise from users' diverse and limited storage, communication, and computational capabilities (Setayesh et al., 2023). Some existing works assume a global model architecture, where each user's local model parameters are a subset of the global model's parameters (Diao et al., 2021; Hong et al., 2022). However, such approaches limit the flexibility for users to choose arbitrary model architectures. To overcome this limitation, one potential solution is to adopt knowledge distillation (KD)-based FL algorithms (Li & Wang, 2019; Park et al., 2024). For example, FCCL (Huang et al., 2022), KT-pFL (Zhang et al., 2021), and FedDF (Lin et al., 2020) leverage KD and an unlabeled public dataset to address model heterogeneity. However, in these approaches, knowledge can only be transferred among models with the same logit size (i.e., models associated with the same task). FedHeNN (Makhija et al., 2022) addresses this issue by introducing a proximal term into users' local loss functions. Centered kernel alignment (CKA) is used in FedHeNN to measure similarity among representation matrices. Nevertheless, the reliability of CKA as a similarity metric is still under investigation (Davari et al., 2023).

**Representation Learning in FL**   CL is a widely used technique for representation learning. Recently, incorporating CL into FL has shown promise in improving the performance of users' models. Federated SimCLR (Louizos et al., 2024) employs the SimCLR loss (Chen et al., 2020) as the users' local loss function. In MOON (Li et al., 2021a), each user applies CL to bring the representations obtained from its local model closer to those of the global model and farther from those of its previous local model. However, both Federated SimCLR and MOON assume homogeneous models across users. FedPCL (Tan et al., 2022) and FedTGP (Zhang et al., 2024) utilize CL with a focus on class-wise prototypes, which makes them inapplicable to multi-task scenarios. multi-modal federated learning (MFL) (Yu et al., 2023; Qi & Li, 2024; Che et al., 2024; Chen et al., 2024) is another line of research that relies on representation learning. CreamFL (Yu et al., 2023) is one of the most effective MFL algorithms, in which local contrastive regularization (LCR) is applied at the users and global-local contrastive aggregation (GCA) is performed at the server to learn a global model. Although the focus of our work is not on obtaining a global model, our experiments show that replacing LCR and GCA in CreamFL with our proposed systematic CL using the Muscle loss improves the performance of the global model on a multi-modal retrieval task.

## 3   PROBLEM FORMULATION

**Notations**   In this paper, we represent vectors by boldface lowercase letters (e.g., $\boldsymbol{z}$) and sets by calligraphic letters (e.g., $\mathcal{Z}$). The cardinality of set $\mathcal{Z}$ and the number of elements in vector $\boldsymbol{z}$ are denoted by $|\mathcal{Z}|$ and $|\boldsymbol{z}|$, respectively. We define $[N] = \{1, 2, \ldots, N\}$ and $\{\boldsymbol{z}_i\}_{i=1}^N = \{\boldsymbol{z}_1, \boldsymbol{z}_2, \ldots \boldsymbol{z}_N\}$. The cosine similarity between two normalized vectors $\boldsymbol{z}_i$ and $\boldsymbol{z}_j$ is given by $\boldsymbol{z}_i \cdot \boldsymbol{z}_j = \boldsymbol{z}_i^T \boldsymbol{z}_j$.

**Problem Setting**   We consider an FL setting with one server and $N$ users, where each user has a local, task-specific labeled dataset. Let $\mathcal{D}^n$ denote the local dataset of user $n \in [N]$. Users may employ heterogeneous models based on their tasks and available resources (e.g., storage capacity and computational capability). Let $\boldsymbol{\theta}^n$ denote the parameters of user $n$'s local model. The optimization problem $\min_{\boldsymbol{\theta}^n} \mathbb{E}[\mathcal{L}^n(\boldsymbol{\theta}^n)]$ can be solved locally by each user $n$ to find a customized model for its task, where $\mathcal{L}^n$ is the loss function corresponding to user $n$'s task, and the expectation is taken over its labeled dataset $\mathcal{D}^n$. However, the key goal of FMTL is to enable users to train their local models under the orchestration of the server, thereby capturing common task-agnostic information. This coordination can improve the generalization capability of the users' models, leading to better performance compared to training solely on their local datasets. Unlike existing FMTL algorithms that employ a parameter-sharing mechanism (e.g., a shared encoder) to capture common information across tasks, we use joint representation learning among the users' models.

For each user $n \in [N]$, we decouple its local model $\boldsymbol{\theta}^n$ into a representation model $\boldsymbol{w}^n$ and a task-specific prediction head $\boldsymbol{\phi}^n$, such that $\boldsymbol{\theta}^n = \{\boldsymbol{w}^n, \boldsymbol{\phi}^n\}$. We assume that the output features of all representation models have the same dimension $d$[1]. We also consider that users have ac-

---

[1]This requirement does not preclude model heterogeneity across users and can be relaxed by appending a lightweight, learnable projection head that maps each user's latent representation to a common dimension.

cess to a shared unlabeled public dataset. This public dataset may be uni-modal or multi-modal, depending on the users' tasks. Moreover, it can be easily obtained by collecting data from relevant domains (Huang et al., 2022), using publicly available datasets (Yu et al., 2023), or generating synthetic data samples (Huang et al., 2024). Let $\mathcal{D}$ denote the shared public dataset. In each communication round of FMTL, each user $n$ first trains its local model $\boldsymbol{\theta}^n$ on its local dataset $\mathcal{D}^n$ for $E$ local epochs. Then, users extract representation vectors corresponding to samples from the public dataset and transmit them to the server. In particular, for each data sample $\boldsymbol{x}_i \in \mathcal{D}$, user $n$ feeds it into $\boldsymbol{w}^n$ to obtain a normalized representation vector denoted by $\boldsymbol{z}_i^n \in \mathbb{R}^d$. Considering a batch of size $B$ from the public dataset, each user $n$ transmits a representation matrix $\boldsymbol{Z}^n \in \mathbb{R}^{B \times d}$ to the server, where each row of $\boldsymbol{Z}^n$ corresponds to a sample in the batch. As illustrated in Figure 1, based on our proposed Muscle loss function (to be presented in Section 4), the server obtains an aggregated matrix and a weighting coefficient vector for each user by using the representation matrices received from the other users. The complete proposed FMTL algorithm is presented in Section 5.

## 4 THE CONTRASTIVE LEARNING FRAMEWORK

In this section, we first explain why pairwise alignment approaches may not fully capture the dependencies among the representations of more than two models. Next, we propose Muscle as a new CL loss and show that it can systematically capture such dependencies. Finally, we theoretically demonstrate the relationship between the Muscle loss and the MI among the models' representations.

### 4.1 PRELIMINARIES

InfoNCE is a popular CL loss function, and its variants have been widely used in the literature to improve the quality of learned representations between two views of the same data (Chen et al., 2020) or across two modalities (Radford et al., 2021). The idea behind the InfoNCE loss function is to learn a representation space that contrasts samples from *two* different distributions. Given $\{\boldsymbol{z}_i^n\}_{i=1}^B$ and $\{\boldsymbol{z}_i^m\}_{i=1}^B$ as the representations corresponding to a batch of $B$ samples, $\boldsymbol{z}_i^n$ and $\boldsymbol{z}_j^m$ are considered positive pairs if $i = j$ (i.e., the representation vectors originate from the same data sample), and negative pairs if $i \neq j$. Without loss of generality, we treat $\boldsymbol{z}_i^n$ as the anchor. That is, we identify its positive and negative counterparts and define the loss function accordingly. The InfoNCE loss is given as follows (Oord et al., 2019):

$$\mathcal{L}_{\text{InfoNCE}}^{n,m}(\boldsymbol{z}_i^n) = -\log \frac{\exp(\boldsymbol{z}_i^n \cdot \boldsymbol{z}_i^m / \tau_{n,m})}{\sum_{j \in [B]} \exp(\boldsymbol{z}_i^n \cdot \boldsymbol{z}_j^m / \tau_{n,m})}, \tag{1}$$

where $\tau_{n,m}$ is a temperature parameter that moderates the effect of similarity. To extend InfoNCE to more than two distributions (e.g., representations obtained from multiple modalities, views, or models), pairwise alignment approaches apply the InfoNCE loss to contrast samples from each pair of distributions (Tian et al., 2020; Yang et al., 2022; Wang & Sun, 2022). In particular, given $\boldsymbol{z}_i^n$ as the anchor, we have $\mathcal{L}_{\text{Pairwise}}^n(\boldsymbol{z}_i^n) = \sum_{m \in [N] \setminus \{n\}} \mathcal{L}_{\text{InfoNCE}}^{n,m}(\boldsymbol{z}_i^n)$. However, since $\mathcal{L}_{\text{Pairwise}}^n(\boldsymbol{z}_i^n)$ accounts only for pairwise dependencies, it cannot effectively capture dependencies among *all* distributions or learn a representation space that contrasts their samples *jointly*.

### 4.2 OUR PROPOSED MUSCLE LOSS FUNCTION

We introduce the Muscle loss, which goes beyond pairwise alignment by jointly considering representations from multiple models. To this end, we focus on the $N$-tuple of representation vectors in learning representations across $N$ models. In particular, all representation vectors $\{\boldsymbol{z}_i^m\}_{m=1,\, m \neq n}^N$ are considered positives for the anchor representation vector $\boldsymbol{z}_i^n$. Also, any combination of representation vectors from all models in $[N] \setminus \{n\}$ (i.e., one from each model except model $n$) is considered a negative for the anchor vector $\boldsymbol{z}_i^n$ if at least one of the vectors in the combination corresponds to a data sample $j \neq i$. We define the Muscle loss function as follows:

$$\mathcal{L}_{\text{Muscle}}^n(\boldsymbol{z}_i^n) = -\log \frac{f(\boldsymbol{z}_i^n, \{\boldsymbol{z}_i^m\}_{m=1,\, m \neq n}^N)}{\sum_{\boldsymbol{j} \in \mathcal{J}^n} f(\boldsymbol{z}_i^n, \{\boldsymbol{z}_{j_m}^m\}_{m=1,\, m \neq n}^N)}, \tag{2}$$

where $f$ is a function that assigns high values to positive $N$-tuples and low values to negative $N$-tuples. For the sake of brevity, we define $\mathcal{J}^n = \{\boldsymbol{j} = (j_1, \dots, j_N) \,|\, j_m \in [B], \, m \in [N] \setminus \{n\}\}$. In Appendix A, we show that the optimal value of $f(\boldsymbol{z}_i^n, \{\boldsymbol{z}_{j_m}^m\}_{m=1,\, m \neq n}^N)$ is proportional to the

probability density ratio $p(z_i^n, \{z_{j_m}^m\}_{m=1, m \neq n}^N)/p(z_i^n)p(\{z_{j_m}^m\}_{m=1, m \neq n}^N)$, and that, the Muscle loss function can be expressed as follows:

$$\mathcal{L}_{\text{Muscle}}^n(z_i^n) = -\log \frac{\alpha_{(i,\ldots,i)} \exp(z_i^n \cdot \sum_{m \in [N] \setminus \{n\}} z_i^m / \tau_{n,m}^{(N)})}{\sum_{j \in \mathcal{J}^n} \alpha_j \exp(z_i^n \cdot \sum_{m \in [N] \setminus \{n\}} z_{j_m}^m / \tau_{n,m}^{(N)})}, \tag{3}$$

where $\tau_{n,m}^{(N)}$ is a temperature parameter that moderates the effect of similarity between the representations of models $n$ and $m$, given the correlations among representations of all $N$ models. Additionally, we have $\alpha_j = \exp(-\frac{1}{2} \sum_{m \in [N] \setminus \{n\}} \sum_{m' \in [N] \setminus \{n,m\}} \gamma_{m,m'}^{(N)} z_{j_m}^m \cdot z_{j_{m'}}^{m'})$, where $\gamma_{m,m'}^{(N)} = 1/\tau_{m,m'}^{(N-1)} - 1/\tau_{m,m'}^{(N)}$. As shown in equation (3), $\alpha_j$, where $j \in \mathcal{J}^n$, serves as a weighting coefficient in the Muscle loss function. These coefficients depend on the similarity among the representations of non-anchor models. In Appendix B, we show that the proposed Muscle loss captures dependencies among representations more effectively than pairwise alignment, owing to the use of these weighting coefficients.

**Remark 1** *In Appendix C, we show that $\tau_{n,m}^{(N)} > \tau_{n,m}^{(N-1)}$, which implies that $\gamma_{m,m'}^{(N)}$ in $\alpha_j$ is always positive. Consequently, greater dissimilarity among the representation vectors $\{z_{j_m}^m\}_{m=1, m \neq n}^N$ results in a larger weighting coefficient $\alpha_j$. This leads the Muscle loss function to place greater emphasis on increasing the dissimilarity between the anchor vector $z_i^n$ and negatives that exhibit a higher dissimilarity among themselves. In Appendix D, we provide a simple example to illustrate the impact of the weighting coefficients $\alpha_j$ in the Muscle loss function.*

The following theorem establishes the relationship between our proposed Muscle loss function and the MI among the models' representations.

**Theorem 1** *Given $\mathcal{L}_{Muscle}^n(z_i^n)$ in equation (3), the mutual information $I(z_i^n; \{z_i^m\}_{m=1, m \neq n}^N)$ is lower-bounded as follows:*

$$I(z_i^n; \{z_i^m\}_{m=1, m \neq n}^N) \geq (N-1)\log(B) - \mathbb{E}\mathcal{L}_{Muscle}^n(z_i^n), \tag{4}$$

*where $\mathbb{E}$ denotes the expectation over random batch of data samples.*

**Proof.** See Appendix E. □

**Remark 2** *Given inequality (4), minimizing the Muscle loss is equivalent to maximizing a lower bound on $I(z_i^n; \{z_i^m\}_{m=1, m \neq n}^N)$. Thus, the Muscle loss facilitates knowledge transfer from models $m \in [N] \setminus \{n\}$ to model $n$ by aligning the representations of model $n$ with those of the other models. In Appendix F, we provide a discussion on why the Muscle loss is more effective for facilitating knowledge transfer among models compared to the pairwise alignment from the MI perspective.*

Our proposed Muscle loss function enables a systematic CL framework by incorporating weighting coefficients $\alpha_j$. In the next section, we demonstrate how this systematic CL framework can be leveraged to develop a novel FMTL algorithm.

## 5 FEDMUSCLE METHODOLOGY

In this section, we propose FedMuscle, a novel FMTL algorithm designed to address both model and task heterogeneity across users. To achieve this, FedMuscle aligns the representation spaces of users' models using the Muscle loss function defined in (3). As discussed in Section 3, given a batch of public data samples, each user $n \in [N]$ sends a representation matrix $Z^n$ to the server. For each user $n$, the server computes an aggregated matrix and a weighting coefficient vector, which are then sent back to the user to compute the CL loss locally as follows:

$$\mathcal{L}_{\text{CL}}^n = -\frac{1}{B} \sum_{i \in [B]} \log \frac{\alpha_{(i,\ldots,i)}^n \exp\left(z_i^n \cdot s_{(i,\ldots,i)}^n\right)}{\sum_{j \in \mathcal{J}^n} \alpha_j^n \exp\left(z_i^n \cdot s_j^n\right)}, \tag{5}$$

where, based on (3), we have $\alpha_j^n = \exp(-\frac{1}{2} \sum_{m \in [N] \setminus \{n\}} \sum_{m' \in [N] \setminus \{n,m\}} \gamma_{m,m'}^{(N)} z_{j_m}^m \cdot z_{j_{m'}}^{m'})$ and $s_j^n = \sum_{m \in [N] \setminus \{n\}} z_{j_m}^m / \tau_{n,m}^{(N)}$. Note that, for each user $n$, the server can compute $\alpha_j^n$ and $s_j^n$ for all $j \in \mathcal{J}^n$ using the representation matrices $Z^m$ received from the other users $m \in [N] \setminus \{n\}$.

---

**Algorithm 1** Training Procedure of FedMuscle

---

1: **Input:** Public dataset $\mathcal{D}$; local dataset $\mathcal{D}^n$, $n \in [N]$; number of local epochs $E$; number of communication rounds $R$; number of CL epochs $T$; batch size $B$; number of selected representation matrices $M$.
2: **Output:** A customized local model $\boldsymbol{\theta}^n$ for each user $n \in [N]$ on its task.
3: Randomly initialize $\boldsymbol{\theta}^n = \{\boldsymbol{w}^n, \boldsymbol{\phi}^n\}$ for each user $n \in [N]$.
4: **for** each communication round $r \in [R]$ **do**
5:     **for** each user $n \in [N]$ in parallel **do**
6:        $\{\boldsymbol{w}^n, \boldsymbol{\phi}^n\} \leftarrow$ **LocalUpdate**$(\boldsymbol{\theta}^n, \mathcal{D}^n, E)$
7:        **for** each CL epoch $t \in [T]$ **do**
8:           **for** each batch of data samples from $\mathcal{D}$ **do**
9:             Obtain the representation matrix $\boldsymbol{Z}^n$ and send it to the server.
10:             Receive $\boldsymbol{S}^n$ and $\boldsymbol{\alpha}^n$ from the server, where $\boldsymbol{S}^n$ and $\boldsymbol{\alpha}^n$ are computed based on the $M$ representation matrices randomly selected by the server for user $n$.
11:             Update $\boldsymbol{w}^n$ by minimizing the CL loss function in (5).
12:           **end for**
13:        **end for**
14:     **end for**
15: **end for**
16: **function** LocalUpdate$(\boldsymbol{\theta}^n, \mathcal{D}^n, E)$
17:     **for** each local epoch $e \in [E]$ **do**
18:        Update $\boldsymbol{\theta}^n$ by minimizing the loss function $\mathcal{L}^n$ corresponding to user $n$'s task on $\mathcal{D}^n$.
19:     **end for**
20: **Return** $\{\boldsymbol{w}^n, \boldsymbol{\phi}^n\}$

---

Let $\boldsymbol{S}^n$ and $\boldsymbol{\alpha}^n$ denote the aggregated matrix and the weighting coefficient vector computed by the server for user $n$, respectively. The rows of $\boldsymbol{S}^n$ and the elements of $\boldsymbol{\alpha}^n$ correspond to $\boldsymbol{s}_{\boldsymbol{j}}^n$ and $\alpha_{\boldsymbol{j}}^n$ for all $\boldsymbol{j} \in \mathcal{J}^n$, respectively. Algorithm 1 summarizes the training procedure of FedMuscle. The users collaboratively train their local models over $R$ communication rounds. In each communication round $r \in [R]$, each user $n \in [N]$ performs $E$ local epochs to update its model $\boldsymbol{\theta}^n$ using its local dataset $\mathcal{D}^n$. The users then perform $T$ CL epochs to update their representation models by aligning the representation spaces of their models. Specifically, in each communication round $r \in [R]$ and each CL epoch $t \in [T]$, each user $n$ sends the representation matrix $\boldsymbol{Z}^n$, obtained from a batch of the public dataset $\mathcal{D}$, to the server and receives the aggregated matrix $\boldsymbol{S}^n$ and the weighting coefficient vector $\boldsymbol{\alpha}^n$ from the server. The user then minimizes the CL loss function $\mathcal{L}_{\text{CL}}^n$ in equation (5).

For each user $n \in [N]$, the communication cost of the FedMuscle algorithm in the uplink direction depends on the size of the representation matrix $\boldsymbol{Z}^n$, which is $B \times d$. The communication cost in the downlink direction, however, depends on the sizes of $\boldsymbol{S}^n$ and $\boldsymbol{\alpha}^n$. If the server uses the representation matrices from all the other users, i.e., $[N] \setminus \{n\}$, to compute $\boldsymbol{S}^n$ and $\boldsymbol{\alpha}^n$, their dimensions are $B^{N-1} \times d$ and $B^{N-1}$, respectively. Note that the large sizes of $\boldsymbol{S}^n$ and $\boldsymbol{\alpha}^n$ may incur high communication costs in the downlink direction, as more parameters need to be transmitted from the server to user $n$, particularly when $N$ is large. To reduce communication costs in FedMuscle, we randomly select only $M$ of the available representation matrices from the $N - 1$ users to compute $\boldsymbol{S}^n$ and $\boldsymbol{\alpha}^n$ for user $n$ in each communication round and CL epoch. Let $\mathcal{N}^n \subseteq [N] \setminus \{n\}$ denote the set of users whose representation matrices are selected to compute the elements of $\boldsymbol{S}^n$ and $\boldsymbol{\alpha}^n$ for user $n$, where $|\mathcal{N}^n| = M$. Thus, we have $\mathcal{J}^n = \{\boldsymbol{j} = (j_m)_{m \in \mathcal{N}^n} \,|\, j_m \in [B], m \in \mathcal{N}^n\}$ to specify the index tuples used in computing the terms of the CL loss function $\mathcal{L}_{\text{CL}}^n$ in equation (5). Consequently, in FedMuscle, we have $\boldsymbol{S}^n \in \mathbb{R}^{|\mathcal{J}^n| \times d}$ and $\boldsymbol{\alpha}^n \in \mathbb{R}^{|\mathcal{J}^n|}$, where $|\mathcal{J}^n| = B^M$.

## 6 EXPERIMENTS

### 6.1 EXPERIMENTAL SETUP

**Datasets and Tasks** We conduct our experiments using datasets such as CIFAR-10, CIFAR-100 (Krizhevsky, 2009), common objects in context (COCO) (Lin et al., 2014), Yahoo! Answers (Zhang et al., 2015), Pascal VOC (Everingham et al., 2015), and Flickr30K (Plummer et al., 2015). Inspired by the experimental setup in (Chen et al., 2020), we consider the following three CV tasks: image classification using CIFAR-10 (IC10), image classification using CIFAR-100 (IC100), and multi-label classification using COCO (MLC). We refer to this uni-modal setup as **Setup1**,

which enables a fair comparison with existing baseline algorithms. To evaluate the performance of FedMuscle in a more challenging, multi-modal setting, we further expand Setup1 by adding two additional tasks: semantic segmentation using COCO (SS) and text classification using Yahoo! Answers (TC). We refer to this multi-modal setup as **Setup2**. For the uni-modal Setup1, we consider that the samples in the public dataset are entirely drawn from one of the following datasets: CIFAR-100, COCO, or Pascal VOC. For the multi-modal Setup2, we consider that the samples in the public dataset are entirely drawn from Flickr30K. In Setup2, users with CV tasks obtain representations from the images in the public dataset, while users with the TC task obtain representations from the image captions. The size of the shared public dataset is set to 5000. For additional details on the number of training and test samples for each task, please refer to Appendix G.

**Implementation Details**   We use heterogeneous FMs as the users' local models. Specifically, our experiments involve heterogeneous ViT models (Dosovitskiy et al., 2021) pre-trained on ImageNet-21K (Deng et al., 2009), heterogeneous SegFormer models (Xie et al., 2021) pre-trained on ImageNet-1K and fine-tuned on ADE20K (Zhou et al., 2017), and heterogeneous BERT-based models (Devlin et al., 2019; Sanh et al., 2019) pre-trained on a large corpus of English text[2]. Details of the selected models and assigned tasks for users in Setup1 and Setup2 are provided in Appendix H. We set $d$ (i.e., the output dimension of the representation models) to 256. To reduce the computational cost for users, we fine-tune their pre-trained FMs using parameter-efficient fine-tuning (PEFT) (Zhang et al., 2023b). Specifically, we use low-rank adaptation (LoRA) (Hu et al., 2022) with a rank of 16. The models are trained using the AdamW optimizer (Loshchilov & Hutter, 2019) with a learning rate of 1e-3. We set the batch size $B$ to 32 and the value of $M$ in Algorithm 1 to 3. The temperature parameters $\tau_{n,m}^{(4)}$ and $\tau_{n,m}^{(3)}$ are set to 0.2 and 0.15, respectively. Unless otherwise stated, the number of local epochs $E$, communication rounds $R$, and CL epochs $T$ are set to 1, 150, and 1, respectively. We implement FedMuscle and the baseline algorithms using PyTorch (Paszke et al., 2019), and run all experiments on NVIDIA Tesla V100 GPUs.

**Baselines**   We compare the performance of our proposed algorithm, FedMuscle, with FedRCL (Seo et al., 2024) as a CL-based FL algorithm; SAGE (Liu et al., 2025) and CoFED (Cao et al., 2023) as FSSL algorithms; FedDF (Lin et al., 2020) as a KD-based FL algorithm; and FedHeNN (Makhija et al., 2022), a model-agnostic FL algorithm. Additionally, we compare FedMuscle with SimCLR (Chen et al., 2020), pseudo-labeling (Cascante-Bonilla et al., 2021), and local training, where each user's model is trained individually. Specifically, SimCLR and pseudo-labeling utilize the public dataset for self-supervised and semi-supervised learning, respectively, while local training relies solely on each user's labeled local dataset.

**Performance Metrics**   For the IC10, IC100, and TC tasks, accuracy on the test samples is used as the performance metric. For the MLC and SS tasks, micro-average F1-score (micro-F1) (Wu & Zhou, 2017) and mean intersection over union (mIoU) (Everingham et al., 2010) are used as the performance metrics, respectively. All the evaluation metrics are expressed as percentages (%), and higher values indicate better performance. Furthermore, to provide an overall evaluation metric for an algorithm's performance, the average per-user performance improvement relative to the local training baseline, denoted as $\Delta$, can be employed (Lu et al., 2024). Specifically, we have $\Delta = \frac{1}{N} \sum_{n \in [N]} (M_{\text{alg}}^n - M_{\text{local}}^n)/M_{\text{local}}^n$, where $M_{\text{alg}}^n$ and $M_{\text{local}}^n$ represent the performance metrics of the considered algorithm and the local training baseline, respectively, for the task of user $n \in [N]$.

## 6.2 Benchmark Experiments

**Comparison with Baselines**   Table 1 presents the performance of FedMuscle compared to the baseline algorithms in the uni-modal Setup1. Based on the results in Table 1, our observations are as follows: (1) The proposed algorithm, FedMuscle, consistently achieves better overall performance in terms of $\Delta$ compared to the baseline algorithms. (2) Using public datasets with detailed images, such as Pascal VOC and COCO, leads to improved performance for FedMuscle[3]. (3) Even when the public dataset is derived from CIFAR-100, which contains less detailed images, FedMuscle still enhances users' performance on their respective tasks. The performance of FedMuscle using a synthetic public dataset is presented in Appendix J. These results confirm that FedMuscle remains

---

[2]Pre-trained FMs are downloaded from Hugging Face at `www.huggingface.co/models`.

[3]In Appendix I, we present a training curve illustrating how a decrease in the Muscle loss corresponds to an increase in $\Delta$ over communication rounds.

Table 1: Performance of FedMuscle compared to the considered baseline algorithms in Setup1.

| Public Dataset | User # | Model | Task | Eval. Metric | Algorithm | | | | | | | | |
|---|---|---|---|---|---|---|---|---|---|---|---|---|---|
| | | | | | FedMuscle (Ours) | SAGE (2025) | FedHeNN (2022) | CoFED (2023) | FedDF (2020) | FedRCL (2024) | SimCLR (2020) | Pseudo-labeling (2021) | Local Training |
| Pascal VOC | 1 | ViT-Base | MLC | micro-F1 | $46.33_{\pm 0.12}$ | $41.97_{\pm 0.34}$ | $41.27_{\pm 0.48}$ | $47.47_{\pm 0.12}$ | $42.43_{\pm 0.17}$ | $41.77_{\pm 0.34}$ | $40.80_{\pm 0.45}$ | $45.93_{\pm 0.52}$ | $42.17_{\pm 0.24}$ |
| | 2 | ViT-Small | MLC | micro-F1 | $49.77_{\pm 0.29}$ | $45.37_{\pm 0.59}$ | $45.87_{\pm 0.46}$ | $48.80_{\pm 0.36}$ | $43.67_{\pm 1.29}$ | $44.07_{\pm 0.29}$ | $44.83_{\pm 0.46}$ | $48.07_{\pm 0.45}$ | $43.67_{\pm 0.59}$ |
| | 3 | ViT-Large | MLC | micro-F1 | $49.40_{\pm 0.50}$ | $44.10_{\pm 0.08}$ | $48.23_{\pm 0.66}$ | $49.77_{\pm 0.49}$ | $42.57_{\pm 0.19}$ | $42.43_{\pm 0.42}$ | $44.53_{\pm 0.68}$ | $48.20_{\pm 0.28}$ | $42.93_{\pm 0.41}$ |
| | 4 | ViT-Base | IC100 | Accuracy | $36.67_{\pm 0.34}$ | $24.50_{\pm 0.57}$ | $24.10_{\pm 0.51}$ | $24.67_{\pm 0.29}$ | $23.93_{\pm 0.41}$ | $25.27_{\pm 0.09}$ | $27.43_{\pm 0.42}$ | $21.40_{\pm 0.33}$ | $24.77_{\pm 0.42}$ |
| | 5 | ViT-Small | IC100 | Accuracy | $29.93_{\pm 0.54}$ | $25.13_{\pm 0.47}$ | $24.83_{\pm 1.10}$ | $23.70_{\pm 1.31}$ | $24.70_{\pm 0.45}$ | $27.23_{\pm 0.61}$ | $23.60_{\pm 0.16}$ | $22.77_{\pm 1.10}$ | $24.70_{\pm 0.36}$ |
| | 6 | ViT-Tiny | IC10 | Accuracy | $66.57_{\pm 1.01}$ | $43.33_{\pm 1.15}$ | $41.63_{\pm 1.33}$ | $43.40_{\pm 0.37}$ | $43.20_{\pm 0.24}$ | $44.63_{\pm 0.12}$ | $49.03_{\pm 0.80}$ | $43.77_{\pm 1.73}$ | $43.77_{\pm 0.62}$ |
| | Δ (%) ↑ | | | | **+26.70** | +0.96 | -0.41 | +5.83 | -0.82 | +2.17 | +3.57 | +1.64 | 0.00 |
| COCO | 1 | ViT-Base | MLC | micro-F1 | $49.10_{\pm 0.45}$ | $41.97_{\pm 0.33}$ | $42.07_{\pm 0.66}$ | $50.87_{\pm 0.38}$ | $43.17_{\pm 0.46}$ | $41.77_{\pm 0.34}$ | $42.50_{\pm 0.70}$ | $47.23_{\pm 0.17}$ | $42.17_{\pm 0.24}$ |
| | 2 | ViT-Small | MLC | micro-F1 | $51.30_{\pm 0.22}$ | $46.97_{\pm 0.17}$ | $46.27_{\pm 0.54}$ | $52.43_{\pm 0.41}$ | $43.67_{\pm 0.54}$ | $44.07_{\pm 0.29}$ | $44.70_{\pm 0.37}$ | $50.30_{\pm 0.24}$ | $43.67_{\pm 0.59}$ |
| | 3 | ViT-Large | MLC | micro-F1 | $50.60_{\pm 0.36}$ | $44.47_{\pm 0.68}$ | $47.17_{\pm 1.77}$ | $53.50_{\pm 0.37}$ | $42.07_{\pm 0.54}$ | $42.43_{\pm 0.42}$ | $45.17_{\pm 1.06}$ | $50.53_{\pm 0.05}$ | $42.93_{\pm 0.41}$ |
| | 4 | ViT-Base | IC100 | Accuracy | $37.27_{\pm 0.78}$ | $25.20_{\pm 0.49}$ | $24.53_{\pm 0.66}$ | $24.83_{\pm 0.17}$ | $24.10_{\pm 0.67}$ | $25.27_{\pm 0.09}$ | $28.87_{\pm 1.11}$ | $21.70_{\pm 0.85}$ | $24.77_{\pm 0.42}$ |
| | 5 | ViT-Small | IC100 | Accuracy | $30.93_{\pm 0.12}$ | $25.47_{\pm 0.52}$ | $25.20_{\pm 0.43}$ | $24.73_{\pm 0.75}$ | $25.23_{\pm 0.33}$ | $27.23_{\pm 0.61}$ | $23.17_{\pm 0.87}$ | $23.67_{\pm 0.66}$ | $24.70_{\pm 0.36}$ |
| | 6 | ViT-Tiny | IC10 | Accuracy | $63.23_{\pm 0.58}$ | $43.03_{\pm 0.54}$ | $43.07_{\pm 1.03}$ | $40.90_{\pm 1.44}$ | $43.20_{\pm 0.24}$ | $44.63_{\pm 0.12}$ | $47.37_{\pm 0.45}$ | $44.13_{\pm 1.73}$ | $43.77_{\pm 0.62}$ |
| | Δ (%) ↑ | | | | **+28.65** | +2.31 | +2.57 | +9.85 | -0.25 | +2.17 | +4.49 | +4.86 | 0.00 |
| CIFAR-100 | 1 | ViT-Base | MLC | micro-F1 | $42.33_{\pm 0.05}$ | $42.10_{\pm 0.14}$ | $41.43_{\pm 1.16}$ | $43.73_{\pm 0.57}$ | $42.40_{\pm 0.22}$ | $41.77_{\pm 0.34}$ | $43.73_{\pm 0.21}$ | $42.17_{\pm 0.24}$ | $42.17_{\pm 0.24}$ |
| | 2 | ViT-Small | MLC | micro-F1 | $46.50_{\pm 0.14}$ | $44.97_{\pm 0.19}$ | $45.90_{\pm 0.83}$ | $47.10_{\pm 0.64}$ | $43.93_{\pm 0.74}$ | $44.07_{\pm 0.29}$ | $43.90_{\pm 0.75}$ | $47.27_{\pm 0.73}$ | $43.67_{\pm 0.59}$ |
| | 3 | ViT-Large | MLC | micro-F1 | $45.63_{\pm 0.68}$ | $44.17_{\pm 0.33}$ | $47.67_{\pm 0.96}$ | $45.23_{\pm 0.49}$ | $43.10_{\pm 0.22}$ | $42.43_{\pm 0.42}$ | $38.33_{\pm 0.17}$ | $46.53_{\pm 0.79}$ | $42.93_{\pm 0.41}$ |
| | 4 | ViT-Base | IC100 | Accuracy | $33.43_{\pm 0.21}$ | $24.67_{\pm 0.34}$ | $24.23_{\pm 1.11}$ | $27.17_{\pm 0.78}$ | $27.17_{\pm 0.74}$ | $25.27_{\pm 0.09}$ | $26.93_{\pm 0.25}$ | $24.77_{\pm 0.42}$ | $24.77_{\pm 0.42}$ |
| | 5 | ViT-Small | IC100 | Accuracy | $29.60_{\pm 0.71}$ | $24.63_{\pm 0.49}$ | $25.50_{\pm 0.24}$ | $27.27_{\pm 1.29}$ | $24.13_{\pm 0.38}$ | $27.23_{\pm 0.61}$ | $22.53_{\pm 1.32}$ | $26.80_{\pm 1.18}$ | $24.70_{\pm 0.36}$ |
| | 6 | ViT-Tiny | IC10 | Accuracy | $58.37_{\pm 0.57}$ | $41.83_{\pm 0.21}$ | $44.43_{\pm 1.99}$ | $43.30_{\pm 0.33}$ | $43.30_{\pm 0.33}$ | $44.63_{\pm 0.12}$ | $50.03_{\pm 1.05}$ | $43.77_{\pm 0.52}$ | $43.77_{\pm 0.62}$ |
| | Δ (%) ↑ | | | | **+16.88** | +0.1 | +2.83 | +5.99 | -1.42 | +2.17 | -4.94 | +6.26 | 0.00 |
| Computation Cost (TeraFLOPS) | | | | | 523 | 1171 | 34 | 586 | 322 | 78 | 849 | 152 | 29 |

effective regardless of the chosen public dataset. However, it can further improve users' overall performance when the public dataset contains feature-rich samples.

**Performance of FedMuscle in Multi-Modal Setup2** Table 2 shows the performance of FedMuscle across a broader range of tasks, including both CV and NLP tasks. The results in Table 2 demonstrate that even with the addition of two users performing the SS task and two users performing the TC task, FedMuscle still improves the overall performance of users on their respective tasks in terms of Δ. Despite the differences in modalities, the use of CL in FedMuscle aligns representations from various modalities into a shared representation space, enabling each user to capture common task-agnostic information from the other users.

Table 2: Performance of FedMuscle in Setup2.

| User # | Model | Task | Eval. Metric | Algorithm | |
|---|---|---|---|---|---|
| | | | | FedMuscle (Ours) | Local Training |
| 1 | ViT-Base | MLC | micro-F1 | $47.80_{\pm 0.40}$ | $42.17_{\pm 0.24}$ |
| 2 | ViT-Small | MLC | micro-F1 | $51.05_{\pm 0.45}$ | $43.67_{\pm 0.59}$ |
| 3 | ViT-Large | MLC | micro-F1 | $49.00_{\pm 0.50}$ | $42.93_{\pm 0.41}$ |
| 4 | ViT-Base | IC100 | Accuracy | $36.15_{\pm 0.25}$ | $24.77_{\pm 0.42}$ |
| 5 | ViT-Small | IC100 | Accuracy | $28.70_{\pm 0.70}$ | $24.70_{\pm 0.36}$ |
| 6 | ViT-Tiny | IC10 | Accuracy | $61.60_{\pm 0.40}$ | $43.77_{\pm 0.62}$ |
| 7 | SegFormer-B0 | SS | mIoU | $33.95_{\pm 1.65}$ | $33.73_{\pm 1.93}$ |
| 8 | SegFormer-B1 | SS | mIoU | $33.40_{\pm 2.20}$ | $32.43_{\pm 0.48}$ |
| 9 | BERT-Base | TC | Accuracy | $45.95_{\pm 0.05}$ | $41.10_{\pm 0.88}$ |
| 10 | DistilBERT-Base | TC | Accuracy | $54.10_{\pm 0.20}$ | $56.03_{\pm 0.98}$ |
| Δ (%) ↑ | | | | **+14.39** | 0.00 |

**Impact of Muscle Loss in FedMuscle** The results in Figure 2 show that, compared to the Gramian-based contrastive loss recently proposed by Cicchetti et al. (2025), the Muscle loss function improves Δ by 11.2%, 28.4%, and 11.1% when Pascal VOC, COCO, and CIFAR-100 are used as the public datasets, respectively[4]. These improvements stem from the fact that the Muscle loss has solid theoretical justifications and can better capture dependencies among the representations obtained from multiple models. In addition, the computation of the Muscle loss is modular and partially offloaded to the server. By contrast, the Gramian-based loss requires computing the determinant of Gramian matrices, which imposes $(M+1)^3$ times higher computational cost on the users. In Appendix L, we provide more details on the features offered by the Muscle loss compared to the Gramian-based loss. Furthermore, a comparison between the Δ values in Figure 2 and Table 1 reveals that even pairwise alignment in FedMuscle yields better performance than baseline algorithms. This highlights the effectiveness of the FedMuscle algorithm under model and task heterogeneity.

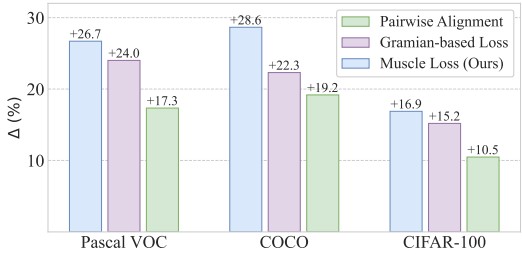

Figure 2: Performance of FedMuscle in Setup1 using the proposed Muscle loss function, compared to the Gramian-based contrastive loss and pairwise alignment.

**Integration of Muscle Loss into a Multi-Modal FL Algorithm** To evaluate the effectiveness of our proposed CL framework, we integrate it into an MFL algorithm. We also aim to demonstrate its effectiveness in a setting with a larger number of users, where the models are trained from scratch under a non-IID partition of local data. To this end, we adopt the CreamFL (Yu et al., 2023)

---

[4]Detailed results can be found in Appendix K.

Table 3: Global model performance on the image-text retrieval task in CreamFL (Yu et al., 2023) setup.

| Eval. Set | Algorithm | i2t_R@1 | i2t_R@5 | i2t_R@10 | t2i_R@1 | t2i_R@5 | t2i_R@10 | Δ (%) ↑ |
|---|---|---|---|---|---|---|---|---|
| 1K Test Images | Local Training | 48.36 | 79.66 | 89.24 | 37.46 | 74.05 | 86.59 | 0.00 |
| | CreamFL | 49.20 | 80.76 | 89.92 | 37.83 | 74.62 | 86.53 | +0.93 |
| | CreamFL + Muscle Loss (Ours) | 49.32 | 80.96 | 90.02 | 38.11 | 74.39 | 86.87 | **+1.17** |
| 5K Test Images | Local Training | 24.78 | 52.44 | 66.30 | 17.72 | 43.28 | 57.47 | 0.00 |
| | CreamFL | 24.48 | 53.48 | 66.92 | 17.96 | 43.74 | 58.15 | +0.88 |
| | CreamFL + Muscle Loss (Ours) | 25.50 | 53.62 | 66.70 | 18.20 | 44.15 | 58.15 | **+1.94** |

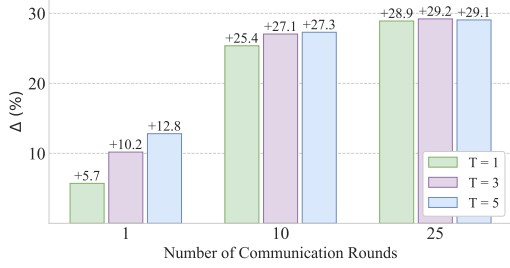

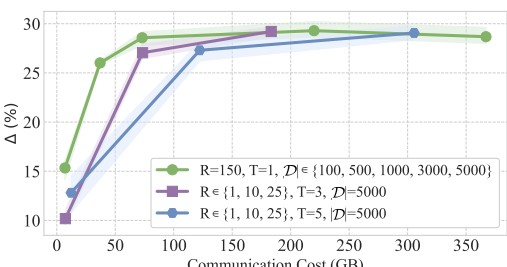

Figure 3: Impact of the number of local epochs $E$, communication rounds $R$, and CL epochs $T$ on Fed-Muscle performance. COCO is used as the public dataset.

Figure 4: Impact of the public dataset size $|\mathcal{D}|$ and number of communication rounds $R$ on the communication cost of FedMuscle. COCO is used as the public dataset.

setup, which consists of 10 uni-modal image users with the CIFAR-100 dataset, 10 uni-modal text users with the AG News dataset (Zhang et al., 2015), and 15 multi-modal users with the Flickr30K dataset. Dirichlet distribution with $\alpha = 0.1$ is used for non-IID data partition. CreamFL aims to learn a global model at the server using representations derived from a public dataset shared among the users and the server. COCO is used as the public dataset. We integrate the Muscle loss into the CreamFL setup by replacing its local contrastive regularization and global-local contrastive aggregation methods with our proposed Muscle loss. More details are provided in Appendix M. Table 3 shows that integrating the Muscle loss into the CreamFL setup improves the global model's performance in terms of image-to-text recall (i2t_R) and text-to-image recall (t2i_R) metrics.

## 6.3 ABLATION STUDIES

**Effect of the Number of Communication Rounds, Local Epochs, and CL Epochs**  Figure 3 illustrates the impact of the number of communication rounds $R$, local epochs $E$, and CL epochs $T$ on the performance of FedMuscle. To ensure a fair comparison, we set $E \times R = 150$ for all cases. Note that higher values of $E$ and $R$ correspond to increased computation and communication costs, respectively. The results in Figure 3 show that a greater number of communication rounds leads to improved performance for FedMuscle[5]. Additionally, when the number of communication rounds is low, FedMuscle can still achieve improved performance by increasing the number of CL epochs.

**Effect of the Public Dataset Size**  Figure 4 illustrates the impact of the public dataset size $|\mathcal{D}|$ on FedMuscle's communication cost and its performance. To provide a comprehensive comparison, we have also included two cases from Figure 3 in Figure 4 (i.e., $T = 3$ and $T = 5$). A larger $|\mathcal{D}|$ leads to higher communication costs as more representation vectors are transmitted between users and the server. However, as shown in Figure 4, FedMuscle is flexible enough to achieve high performance in terms of $\Delta$ while maintaining a low communication cost by appropriately adjusting the number of communication rounds $R$, the number of CL epochs $T$, and the public dataset size $|\mathcal{D}|$.

**Effect of Non-IID Data Distribution and Number of Users**  We consider a setup where the users can perform one of the following four tasks: IC100, IC10, image classification using Food-101 (Bossard et al., 2014) (ICF), and image classification using Caltech-101 (Fei-Fei et al., 2004) (ICC). Specifically, users 1–3 perform IC100, users 4–6 perform IC10, users 7–9 perform ICF, and users 10–12 perform ICC. For those performing the same task, the local data samples are divided across them using a Dirichlet distribution with $\alpha = 0.1$. All users have 100 training samples and 2000 test samples. In Appendix O, we have illustrated the non-IID data partitioning across users.

---

[5] Detailed results can be found in Appendix N.

Table 4: Performance of FedMuscle in the non-IID setting, where users assigned to the same task have heterogeneous local data distributions. The public dataset is derived from Pascal VOC.

| Algorithm | $N$ | User # | | | | | | | | | | | | $\Delta$ (%) $\uparrow$ |
|---|---|---|---|---|---|---|---|---|---|---|---|---|---|---|
| | | 1 | 2 | 3 | 4 | 5 | 6 | 7 | 8 | 9 | 10 | 11 | 12 | |
| Local Training | - | 45.2 | 54.2 | 30.8 | 94.2 | 94.8 | 79.2 | 40.0 | 36.2 | 13.6 | 78.7 | 75.8 | 62.6 | 0.00 |
| FedMuscle (Ours) | 4 | 58.4 | - | - | 94.8 | - | - | 49.9 | - | - | 83.4 | - | - | +15.14 |
| | 8 | 58.0 | 55.4 | - | 94.2 | 93.7 | - | 50.3 | 42.5 | - | 82.4 | 79.0 | - | +10.18 |
| | 12 | 57.7 | 54.2 | 41.4 | 95.1 | 92.8 | 86.0 | 47.8 | 40.3 | 25.6 | 81.8 | 78.0 | 71.0 | +17.40 |

Furthermore, we consider heterogeneous models: users 1, 4, 7, and 10 use ViT-Base, users 2, 5, 8, and 11 use ViT-Small, and users 3, 6, 9, and 12 use ViT-Tiny.

Table 4 presents the performance of FedMuscle under the aforementioned non-IID data partitioning across users for three scenarios, where the number of users is set to 4, 8, and 12. The results in Table 4 show that FedMuscle improves the performance of each user on its respective task. Thus, the overall performance, measured in terms of $\Delta$, is improved by FedMuscle regardless of the data, task, and model heterogeneity among users, and the number of users in the system.

**Effect of Number of Selected Representation Matrices** We present an ablation study on the number of selected representation matrices $M$ in Table 5. The results are obtained under Setup1, using Pascal VOC as the public dataset. As shown in the table, increasing $M$ improves overall performance in terms of $\Delta$, but also increases the communication cost per round. Notably, the performance gains from increasing $M$ beyond 3 (e.g., from 3 to 4 or 5) are marginal. Therefore, we adopt $M = 3$ with random user selection in our experiments, as this offers a balanced trade-off between performance and communication overhead. We also find that random selection provides sufficient diversity across rounds to facilitate knowledge transfer among all tasks.

Table 5: Performance of FedMuscle under different number of selected representation matrices $M$.

| $M$ | 1 | 2 | 3 | 4 | 5 |
|---|---|---|---|---|---|
| $\Delta$ | +17.90 | +24.17 | +26.70 | +27.53 | +27.74 |
| Communication Cost per Round (GB) | 0.004 | 0.050 | 0.956 | 19.080 | 381.565 |

**Effect of Batch Size** We investigate the impact of batch size $B$ on FedMuscle's performance in Setup1 when COCO is used as the public dataset. Note that by increasing the batch size, the approximation used for deriving inequality (23) in Appendix E becomes more accurate. Thus, based on Theorem 1, minimizing the Muscle loss provides a tighter lower bound on maximizing the MI between the models' outputs. The results in Figure 5 also show that by increasing the batch size, the performance of FedMuscle can be improved in terms of the overall performance metric $\Delta$. However, based on the provided discussion in Section 5, we know that larger batch sizes can incur higher communication costs in the downlink direction. To achieve a balanced trade-off between performance and communication overhead, we adopt $B = 32$ in our experiments. Additional ablation studies on the output dimension of the representation models $d$ and temperature parameters $\tau_{n,m}^{(4)}$ and $\tau_{n,m}^{(3)}$ are presented in Appendix P.

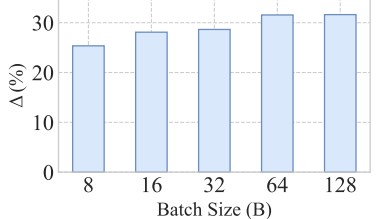

Figure 5: Impact of batch size on FedMuscle's performance in Setup1. COCO is used as the public dataset.

## 7 CONCLUSION

In this work, we proposed a new CL loss function, called Muscle, which systematically and effectively captures dependencies among representations obtained from multiple models. These dependencies are modeled through theoretically grounded weighting coefficients in the Muscle loss. Furthermore, we showed that minimizing the Muscle loss is equivalent to maximizing a lower bound on the MI between the models' representations. Building on the Muscle loss, we designed FedMuscle, a novel and effective FMTL algorithm that enables collaborative training of heterogeneous models in the presence of task heterogeneity across users. To this end, FedMuscle aligns the representation spaces of users' models using a shared public dataset to capture common, task-agnostic information. Through extensive experiments, we showed that FedMuscle consistently improves users' performance on their respective tasks compared to state-of-the-art baseline algorithms under both model and task heterogeneity. A discussion of possible future work is provided in Appendix Q.

REPRODUCIBILITY STATEMENT

For theoretical results, we provide the complete proofs along with their underlying assumptions in Section 4 and Appendices A, B, C, E, F, and S. All data used in this study are publicly available. The code for running our algorithm is also provided in the supplementary material. Furthermore, we provide all experimental details in Section 6 and Appendices G, H, and M. We have cited all original sources for the models, baselines, and datasets used in this work. Please refer to Section 6 for further details.

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

We provide more details and results about our work in the appendices. Here are the contents:

## A   DERIVATION OF OUR PROPOSED MUSCLE LOSS FUNCTION

Considering equation (2), we have

$$\mathbb{E}\mathcal{L}_{\text{Muscle}}^{n}(\boldsymbol{z}_i^n) = -\mathbb{E}\log \frac{f(\boldsymbol{z}_i^n, \{\boldsymbol{z}_i^m\}_{m=1,\,m\neq n}^{N})}{\sum_{\boldsymbol{j}\in\mathcal{J}^n} f(\boldsymbol{z}_i^n, \{\boldsymbol{z}_{j_m}^m\}_{m=1,\,m\neq n}^{N})}, \tag{6}$$

where $\mathbb{E}$ denotes the expectation over random batch of data samples. Equation (6) can be interpreted as the categorical cross-entropy for correctly classifying the positives, where the fraction inside the $\log(\cdot)$ function represents the predicted probability. The optimal expected categorical cross-entropy is obtained as $-\mathbb{E}\log p(\text{correctly classifying positives})$, where

$p(\text{correctly classifying positives}) =$

$$\frac{p\left(\{\boldsymbol{z}_i^m\}_{m=1,\,m\neq n}^{N} \,\big|\, \boldsymbol{z}_i^n\right) \prod_{\boldsymbol{j}'\in\mathcal{J}^n\setminus(i,\dots,i)} p\left(\{\boldsymbol{z}_{j_m'}^m\}_{m=1,\,m\neq n}^{N}\right)}{\sum_{\boldsymbol{j}\in\mathcal{J}^n} p\left(\{\boldsymbol{z}_{j_m}^m\}_{m=1,\,m\neq n}^{N} \,\big|\, \boldsymbol{z}_i^n\right) \prod_{\boldsymbol{j}'\in\mathcal{J}^n\setminus(j_1,\dots,j_N)} p\left(\{\boldsymbol{z}_{j_m'}^m\}_{m=1,\,m\neq n}^{N}\right)} \tag{7}$$

$$= \frac{\frac{p\left(\{\boldsymbol{z}_i^m\}_{m=1,\,m\neq n}^{N} \,\big|\, \boldsymbol{z}_i^n\right)}{p\left(\{\boldsymbol{z}_i^m\}_{m=1,\,m\neq n}^{N}\right)}}{\sum_{\boldsymbol{j}\in\mathcal{J}^n} \frac{p\left(\{\boldsymbol{z}_{j_m}^m\}_{m=1,\,m\neq n}^{N} \,\big|\, \boldsymbol{z}_i^n\right)}{p\left(\{\boldsymbol{z}_{j_m}^m\}_{m=1,\,m\neq n}^{N}\right)}}. \tag{8}$$

By comparing equations (6) and (8), we observe that the optimal value of $f\left(z_i^n, \{z_{j_m}^m\}_{m=1,\,m\neq n}^N\right)$ is obtained as follows:

$$f^*\left(z_i^n, \{z_{j_m}^m\}_{m=1,\,m\neq n}^N\right) \propto \frac{p\left(\{z_{j_m}^m\}_{m=1,\,m\neq n}^N \mid z_i^n\right)}{p\left(\{z_{j_m}^m\}_{m=1,\,m\neq n}^N\right)} \tag{9}$$

$$= \frac{p\left(z_i^n, \{z_{j_m}^m\}_{m=1,\,m\neq n}^N\right)}{p\left(z_i^n\right)p\left(\{z_{j_m}^m\}_{m=1,\,m\neq n}^N\right)}, \tag{10}$$

where $\propto$ denotes proportionality, implying equivalence up to a multiplicative constant.

In the InfoNCE loss, $f^*$ is associated with the cosine similarity between two vectors (i.e., positive pairs). However, as inferred from equation (10), in our proposed Muscle loss, $f^*$ should be related to the similarity of more than two vectors. To this end, we define a cosine similarity matrix, each element of which is the cosine similarity between two vectors.

Let $C \in \mathbb{R}^{N \times N}$ denote the cosine similarity matrix corresponding to the set of vectors $\{u_k \in \mathbb{R}^d \mid k \in [N]\}$, where $d$ is the dimension of each vector in the set. We have $C_{k,l} = u_k \cdot u_l$, $k, l \in [N]$. Using matrix $C$, we consider that the following expression holds:

$$\frac{p\left(\{u_k\}_{k=1}^N\right)}{\prod_{k\in[N]} p\left(u_k\right)} \propto \exp\left(\frac{1}{2}\sum_{k\in[N]}\sum_{l\in[N]\setminus\{k\}} C_{k,l}/\tau_{k,l}^{(N)}\right) \tag{11}$$

$$= \exp\left(\frac{1}{2}\sum_{k\in[N]}\sum_{l\in[N]\setminus\{k\}} u_k \cdot u_l/\tau_{k,l}^{(N)}\right), \tag{12}$$

where $\tau_{k,l}^{(N)} = \tau_{l,k}^{(N)}$ is a temperature parameter that moderates the effect of similarity between vectors $u_k$ and $u_l$, given the correlations among all $N$ vectors. Note that if we set $N = 2$ in expression (12), it simplifies to $\frac{p(u_k,u_l)}{p(u_k)p(u_l)} \propto \exp\left(u_k \cdot u_l/\tau_{k,l}^{(2)}\right)$. Since there are no additional vectors influencing the similarity between $u_k$ and $u_l$ when $N = 2$, we can set $\tau_{k,l}^{(2)} = \tau_{k,l}$, where $\tau_{k,l}$ is the temperature parameter used in the InfoNCE loss. Therefore, for $N = 2$, $\frac{p(u_k,u_l)}{p(u_k)p(u_l)} \propto \exp\left(u_k \cdot u_l/\tau_{k,l}\right)$, which aligns with the formulation used in the InfoNCE loss (Oord et al., 2019; Tian et al., 2020). Furthermore, for a set with more than two vectors, expression (12) accounts for interactions among all vector pairs by incorporating cosine similarity across all pairs, along with distinct temperature parameters.

Based on expression (12), the optimal function $f^*$ in (10) is obtained as follows:

$$f^*\left(z_i^n, \{z_{j_m}^m\}_{m=1,\,m\neq n}^N\right) \propto$$
$$\frac{\exp\left(\frac{1}{2}\left(2\sum_{m\in[N]\setminus\{n\}} z_i^n \cdot z_{j_m}^m/\tau_{n,m}^{(N)} + \sum_{m\in[N]\setminus\{n\}}\sum_{m'\in[N]\setminus\{n,m\}} z_{j_m}^m \cdot z_{j_{m'}}^{m'}/\tau_{m,m'}^{(N)}\right)\right)}{\exp\left(\frac{1}{2}\sum_{m\in[N]\setminus\{n\}}\sum_{m'\in[N]\setminus\{n,m\}} z_{j_m}^m \cdot z_{j_{m'}}^{m'}/\tau_{m,m'}^{(N-1)}\right)} \tag{13}$$

$$= \frac{\exp\left(\sum_{m\in[N]\setminus\{n\}} z_i^n \cdot z_{j_m}^m/\tau_{n,m}^{(N)}\right)}{\exp\left(\frac{1}{2}\sum_{m\in[N]\setminus\{n\}}\sum_{m'\in[N]\setminus\{n,m\}}(1/\tau_{m,m'}^{(N-1)} - 1/\tau_{m,m'}^{(N)})z_{j_m}^m \cdot z_{j_{m'}}^{m'}\right)} \tag{14}$$

$$= \alpha_j \exp\left(z_i^n \cdot \sum_{m\in[N]\setminus\{n\}} z_{j_m}^m/\tau_{n,m}^{(N)}\right), \tag{15}$$

where $\alpha_j = \exp\left(-\frac{1}{2}\sum_{m\in[N]\setminus\{n\}}\sum_{m'\in[N]\setminus\{n,m\}} \gamma_{m,m'}^{(N)} z_{j_m}^m \cdot z_{j_{m'}}^{m'}\right)$, and $\gamma_{m,m'}^{(N)} = 1/\tau_{m,m'}^{(N-1)} - 1/\tau_{m,m'}^{(N)}$. Combining (15) and (2) results in the Muscle loss function $\mathcal{L}_{\text{Muscle}}^n(z_i^n)$ defined in equation (3).

# B  EFFECT OF $\alpha_j$ IN CAPTURING DEPENDENCIES

In this section, we show that, compared to pairwise alignment (Tian et al., 2020; Wang & Sun, 2022; Yang et al., 2022; Xue et al., 2024a), our proposed Muscle loss more effectively captures dependencies among representations obtained from more than two models. This improvement arises from the use of weighting coefficients $\alpha_j$, $j \in \mathcal{J}^n$.

We consider a special case in which there is no dependency among the representation vectors. In this case, $\alpha_j = 1$ for all $j \in \mathcal{J}^n$. Thus, we have

$$
\begin{aligned}
\mathcal{L}^n_{\text{Muscle}}(\boldsymbol{z}^n_i) &= -\log \frac{\alpha_{(i,\ldots,i)} \exp(\boldsymbol{z}^n_i \cdot \sum_{m\in[N]\setminus\{n\}} \boldsymbol{z}^m_i / \tau^{(N)}_{n,m})}{\sum_{\boldsymbol{j}\in\mathcal{J}^n} \alpha_{\boldsymbol{j}} \exp(\boldsymbol{z}^n_i \cdot \sum_{m\in[N]\setminus\{n\}} \boldsymbol{z}^m_{j_m} / \tau^{(N)}_{n,m})} \\
&\stackrel{(a)}{=} -\log \frac{\exp(\boldsymbol{z}^n_i \cdot \sum_{m\in[N]\setminus\{n\}} \boldsymbol{z}^m_i / \tau^{(N)}_{n,m})}{\sum_{\boldsymbol{j}\in\mathcal{J}^n} \exp(\boldsymbol{z}^n_i \cdot \sum_{m\in[N]\setminus\{n\}} \boldsymbol{z}^m_{j_m} / \tau^{(N)}_{n,m})} \\
&\stackrel{(b)}{=} -\sum_{m\in[N]\setminus\{n\}} \log \frac{\exp(\boldsymbol{z}^n_i \cdot \boldsymbol{z}^m_i / \tau^{(N)}_{n,m})}{\sum_{j_m\in[B]} \exp(\boldsymbol{z}^n_i \cdot \boldsymbol{z}^m_{j_m} / \tau^{(N)}_{n,m})} \\
&\stackrel{(c)}{=} \sum_{m\in[N]\setminus\{n\}} \mathcal{L}^{n,m}_{\text{InfoNCE}}(\boldsymbol{z}^n_i) \\
&\stackrel{(d)}{=} \mathcal{L}^n_{\text{Pairwise}}(\boldsymbol{z}^n_i),
\end{aligned}
\tag{16}
$$

where equality (a) follows from the assumption that $\alpha_j = 1$, $j \in \mathcal{J}^n$; equality (b) follows from the logarithmic product rule; and equalities (c) and (d) follow from the InfoNCE and pairwise alignment formulations provided in Section 4.1.

Equality (16) shows that, when there is no dependency among the representation vectors, our proposed Muscle loss and pairwise alignment are equivalent. Nevertheless, the assumption of no dependency among representation vectors is generally incorrect, as some representation vectors may originate from the same data sample. While pairwise alignment approaches inherently rely on this assumption, our proposed Muscle loss can effectively capture dependencies among representation vectors through the use of weighting coefficients $\alpha_j$, $j \in \mathcal{J}^n$.

# C  THE RELATION BETWEEN $\tau_{k,l}^{(N)}$ AND $\tau_{k,l}^{(N-1)}$

As discussed in Appendix A, the temperature parameter $\tau_{k,l}^{(N)}$ moderates the effect of similarity between vectors $\boldsymbol{u}_k$ and $\boldsymbol{u}_l$ when modeling the probability density ratio among $N$ vectors in (12). In this section, we show that the temperature parameter $\tau_{k,l}^{(N-1)}$, which serves the same purpose for $N-1$ vectors, should be smaller than $\tau_{k,l}^{(N)}$. From (12), for $N$ vectors, we have

$$\frac{p\left(\{\boldsymbol{u}_k\}_{k=1}^{N}\right)}{\prod_{k\in[N]} p\left(\boldsymbol{u}_k\right)} \propto \exp\left(\frac{1}{2}\sum_{k=1}^{N}\sum_{l\in[N]\setminus\{k\}} \boldsymbol{u}_k\cdot\boldsymbol{u}_l/\tau_{k,l}^{(N)}\right). \tag{17}$$

Similarly, for $N-1$ vectors, we have

$$\frac{p\left(\{\boldsymbol{u}_k\}_{k=1}^{N-1}\right)}{\prod_{k\in[N-1]} p\left(\boldsymbol{u}_k\right)} \propto \exp\left(\frac{1}{2}\sum_{k=1}^{N-1}\sum_{l\in[N-1]\setminus\{k\}} \boldsymbol{u}_k\cdot\boldsymbol{u}_l/\tau_{k,l}^{(N-1)}\right). \tag{18}$$

Now, we aim to derive an expression similar to (18) using (17). We have

$$\frac{p\left(\{\boldsymbol{u}_k\}_{k=1}^{N-1}\right)}{\prod_{k\in[N-1]} p\left(\boldsymbol{u}_k\right)} = \int_{\mathbb{R}^d} \frac{p\left(\{\boldsymbol{u}_k\}_{k=1}^{N}\right)}{\prod_{k\in[N]} p\left(\boldsymbol{u}_k\right)} p\left(\boldsymbol{u}_N\right)\, d\boldsymbol{u}_N$$

$$\stackrel{(a)}{\propto} \int_{\mathbb{R}^d} \exp\left(\frac{1}{2}\sum_{k\in[N]}\sum_{l\in[N]\setminus\{k\}} \boldsymbol{u}_k\cdot\boldsymbol{u}_l/\tau_{k,l}^{(N)}\right) p\left(\boldsymbol{u}_N\right)\, d\boldsymbol{u}_N$$

$$= \int_{\mathbb{R}^d} \exp\left(\frac{1}{2}\sum_{k=1}^{N-1}\sum_{l\in[N-1]\setminus\{k\}} \boldsymbol{u}_k\cdot\boldsymbol{u}_l/\tau_{k,l}^{(N)} + \sum_{k=1}^{N-1} \boldsymbol{u}_k\cdot\boldsymbol{u}_N/\tau_{k,N}^{(N)}\right) p\left(\boldsymbol{u}_N\right)\, d\boldsymbol{u}_N$$

$$= \exp\left(\frac{1}{2}\sum_{k=1}^{N-1}\sum_{l\in[N-1]\setminus\{k\}} \boldsymbol{u}_k\cdot\boldsymbol{u}_l/\tau_{k,l}^{(N)}\right) \underbrace{\int_{\mathbb{R}^d} \exp\left(\sum_{k=1}^{N-1} \boldsymbol{u}_k\cdot\boldsymbol{u}_N/\tau_{k,N}^{(N)}\right) p\left(\boldsymbol{u}_N\right)\, d\boldsymbol{u}_N}_{A}, \tag{19}$$

where (a) results from (17).

Next, we aim to find a simplified expression for $A$ in (19). We have

$$A = \int_{\mathbb{R}^d} \exp\left(\boldsymbol{u}_N\cdot\sum_{k=1}^{N-1} \boldsymbol{u}_k/\tau_{k,N}^{(N)}\right) p\left(\boldsymbol{u}_N\right)\, d\boldsymbol{u}_N$$

$$\stackrel{(a)}{\propto} \int_{\mathbb{R}^d} \exp\left(\boldsymbol{u}_N\cdot\sum_{k=1}^{N-1} \boldsymbol{u}_k/\tau_{k,N}^{(N)} - \frac{1}{2}\|\boldsymbol{u}_N\|^2\right) d\boldsymbol{u}_N$$

$$= \int_{\mathbb{R}^d} \exp\left(\frac{1}{2}\left\|\sum_{k=1}^{N-1} \boldsymbol{u}_k/\tau_{k,N}^{(N)}\right\|^2 - \frac{1}{2}\left\|\boldsymbol{u}_N - \sum_{k=1}^{N-1} \boldsymbol{u}_k/\tau_{k,N}^{(N)}\right\|^2\right) d\boldsymbol{u}_N$$

$$= \exp\left(\frac{1}{2}\left\|\sum_{k=1}^{N-1} \boldsymbol{u}_k/\tau_{k,N}^{(N)}\right\|^2\right) \int_{\mathbb{R}^d} \exp\left(-\frac{1}{2}\left\|\boldsymbol{u}_N - \sum_{k=1}^{N-1} \boldsymbol{u}_k/\tau_{k,N}^{(N)}\right\|^2\right) d\boldsymbol{u}_N$$

$$\stackrel{(b)}{\propto} \exp\left(\frac{1}{2}\sum_{k=1}^{N-1}\sum_{l=1}^{N-1} \boldsymbol{u}_k\cdot\boldsymbol{u}_l/\tau_{k,N}^{(N)}\tau_{l,N}^{(N)}\right), \tag{20}$$

where to obtain (a), we assume that the marginal distribution of each $\boldsymbol{u}_k$, for $k\in[N]$, is an isotropic Gaussian. Thus, we have $p\left(\boldsymbol{u}_N\right) \propto \exp\left(-\frac{1}{2}\|\boldsymbol{u}_N\|^2\right)$. Also, (b) follows from the fact that the integral $\int_{\mathbb{R}^d} \exp\left(-\frac{1}{2}\left\|\boldsymbol{u}_N - \sum_{k=1}^{N-1} \boldsymbol{u}_k/\tau_{k,N}^{(N)}\right\|^2\right) d\boldsymbol{u}_N$ evaluates to a constant.

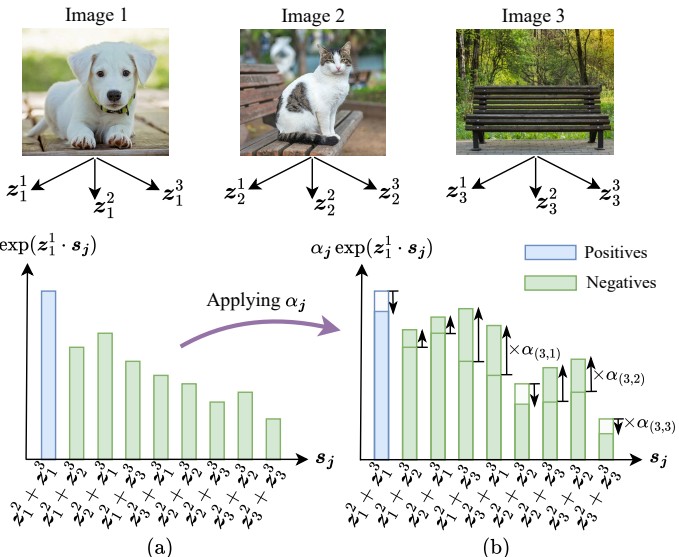

Figure 6: Illustration of the impact of $\alpha_{\boldsymbol{j}}$, $\boldsymbol{j} \in \mathcal{J}^n$ on different terms of our proposed Muscle loss function in (3) for a batch of three images: (a) when $\alpha_{\boldsymbol{j}} = 1$, and (b) when $\alpha_{\boldsymbol{j}}$ is determined based on the dependencies among outputs of representation models. For ease of visualization, $\tau_{n,m}^{(3)}$ is set to one for $n, m \in \{1, 2, 3\}$.

By combining (19) and (20), we have

$$\frac{p\left(\{\boldsymbol{u}_k\}_{k=1}^{N-1}\right)}{\prod_{k \in [N-1]} p\left(\boldsymbol{u}_k\right)} \propto \exp\left(\frac{1}{2}\sum_{k=1}^{N-1}\sum_{l \in [N-1] \setminus \{k\}} \boldsymbol{u}_k \cdot \boldsymbol{u}_l \left(1/\tau_{k,l}^{(N)} + 1/\tau_{k,N}^{(N)}\tau_{l,N}^{(N)}\right)\right). \qquad (21)$$

Comparing (18) with (21) shows that the temperature parameters must satisfy the following equation:

$$1/\tau_{k,l}^{(N-1)} = 1/\tau_{k,l}^{(N)} + 1/\tau_{k,N}^{(N)}\tau_{l,N}^{(N)}, \quad l, k \in [N-1] \qquad (22)$$

Since temperature parameters are positive, equality (22) implies that the temperature parameters must satisfy the following inequality: $\tau_{k,l}^{(N-1)} < \tau_{k,l}^{(N)}$.

## D    EFFECT OF THE WEIGHTING COEFFICIENT $\alpha_{\boldsymbol{j}}$ IN A TOY EXAMPLE

Figure 6 illustrates how $\alpha_{\boldsymbol{j}}$, $\boldsymbol{j} \in \mathcal{J}^n$, impacts the terms in our proposed Muscle loss function. Consider a batch of three images consisting of a dog (image 1), a cat (image 2), and a bench (image 3). The images are selected such that images 1 and 2 are more similar than images 2 and 3, and images 1 and 3 are the least similar. Figure 6(a) and (b) correspond to the cases where $\alpha_{\boldsymbol{j}} = 1$, $\boldsymbol{j} \in \mathcal{J}^n$ (i.e., similar to the pairwise alignment) and where $\alpha_{\boldsymbol{j}}$, $\boldsymbol{j} \in \mathcal{J}^n$, are obtained based on our proposed Muscle loss function in (3), respectively. $\boldsymbol{z}_1^1$ is the output of the representation model 1 for image 1. When $\boldsymbol{z}_1^1$ is considered as the anchor vector, Figure 6(a) shows that the terms corresponding to the negatives originating from images 1 and 2 (e.g., $\boldsymbol{z}_2^2 + \boldsymbol{z}_1^3$) have a greater impact on the loss function when $\alpha_{\boldsymbol{j}} = 1$, $\boldsymbol{j} \in \mathcal{J}^n$. However, this is not the case in Figure 6(b). Specifically, $\alpha_{\boldsymbol{j}}$ tends to have a higher value when the negatives originate from images 1 and 3 (e.g., $\boldsymbol{z}_1^2 + \boldsymbol{z}_3^3$), which are more dissimilar. Therefore, in the Muscle loss function, $\alpha_{\boldsymbol{j}}$ can adjust the importance of the terms in the loss function based on the dissimilarity among the negatives.

# E  PROOF OF THEOREM 1

Based on equation (6), we have

$$\mathbb{E}\mathcal{L}_{\text{Muscle}}^n(\boldsymbol{z}_i^n) = -\mathbb{E}\log\frac{f(\boldsymbol{z}_i^n, \{\boldsymbol{z}_i^m\}_{m=1,\,m\neq n}^N)}{\sum_{\boldsymbol{j}\in\mathcal{J}^n}f(\boldsymbol{z}_i^n, \{\boldsymbol{z}_{j_m}^m\}_{m=1,\,m\neq n}^N)}$$

$$\overset{(a)}{=} -\mathbb{E}\log\left(\frac{\frac{p(\boldsymbol{z}_i^n, \{\boldsymbol{z}_i^m\}_{m=1,\,m\neq n}^N)}{p(\boldsymbol{z}_i^n)p(\{\boldsymbol{z}_i^m\}_{m=1,\,m\neq n}^N)}}{\sum_{\boldsymbol{j}\in\mathcal{J}^n}\frac{p(\boldsymbol{z}_i^n, \{\boldsymbol{z}_{j_m}^m\}_{m=1,\,m\neq n}^N)}{p(\boldsymbol{z}_i^n)p(\{\boldsymbol{z}_{j_m}^m\}_{m=1,\,m\neq n}^N)}}\right)$$

$$= -\mathbb{E}\log\left(\frac{\frac{p(\boldsymbol{z}_i^n, \{\boldsymbol{z}_i^m\}_{m=1,\,m\neq n}^N)}{p(\boldsymbol{z}_i^n)p(\{\boldsymbol{z}_i^m\}_{m=1,\,m\neq n}^N)}}{\frac{p(\boldsymbol{z}_i^n, \{\boldsymbol{z}_i^m\}_{m=1,\,m\neq n}^N)}{p(\boldsymbol{z}_i^n)p(\{\boldsymbol{z}_i^m\}_{m=1,\,m\neq n}^N)} + \sum_{\boldsymbol{j}\in\mathcal{J}^n\backslash(i,\ldots,i)}\frac{p(\boldsymbol{z}_i^n, \{\boldsymbol{z}_{j_m}^m\}_{m=1,\,m\neq n}^N)}{p(\boldsymbol{z}_i^n)p(\{\boldsymbol{z}_{j_m}^m\}_{m=1,\,m\neq n}^N)}}\right)$$

$$= \mathbb{E}\log\left(1 + \frac{p(\boldsymbol{z}_i^n)\,p\left(\{\boldsymbol{z}_i^m\}_{m=1,\,m\neq n}^N\right)}{p\left(\boldsymbol{z}_i^n, \{\boldsymbol{z}_i^m\}_{m=1,\,m\neq n}^N\right)}\sum_{\boldsymbol{j}\in\mathcal{J}^n\backslash(i,\ldots,i)}\frac{p\left(\boldsymbol{z}_i^n, \{\boldsymbol{z}_{j_m}^m\}_{m=1,\,m\neq n}^N\right)}{p(\boldsymbol{z}_i^n)\,p\left(\{\boldsymbol{z}_{j_m}^m\}_{m=1,\,m\neq n}^N\right)}\right)$$

$$\approx \mathbb{E}\log\left(1 + \frac{p(\boldsymbol{z}_i^n)\,p\left(\{\boldsymbol{z}_i^m\}_{m=1,\,m\neq n}^N\right)}{p\left(\boldsymbol{z}_i^n, \{\boldsymbol{z}_i^m\}_{m=1,\,m\neq n}^N\right)}|\mathcal{J}^n|\,\mathbb{E}\frac{p\left(\{\boldsymbol{z}_{j_m}^m\}_{m=1,\,m\neq n}^N\mid \boldsymbol{z}_i^n\right)}{p\left(\{\boldsymbol{z}_{j_m}^m\}_{m=1,\,m\neq n}^N\right)}\right)$$

$$\overset{(b)}{=} \mathbb{E}\log\left(1 + \frac{p(\boldsymbol{z}_i^n)\,p\left(\{\boldsymbol{z}_i^m\}_{m=1,\,m\neq n}^N\right)}{p\left(\boldsymbol{z}_i^n, \{\boldsymbol{z}_i^m\}_{m=1,\,m\neq n}^N\right)}B^{N-1}\right)$$

$$\geq (N-1)\log(B) - \mathbb{E}\log\left(\frac{p\left(\boldsymbol{z}_i^n, \{\boldsymbol{z}_i^m\}_{m=1,\,m\neq n}^N\right)}{p(\boldsymbol{z}_i^n)\,p\left(\{\boldsymbol{z}_i^m\}_{m=1,\,m\neq n}^N\right)}\right)$$

$$\overset{(c)}{=} (N-1)\log(B) - I(\boldsymbol{z}_i^n; \{\boldsymbol{z}_i^m\}_{m=1,\,m\neq n}^N), \tag{23}$$

where equality (a) results from (10). Equality (b) is obtained by using $|\mathcal{J}^n| = B^{N-1}$. Inequality (c) results from the definition of $I(\boldsymbol{z}_i^n; \{\boldsymbol{z}_i^m\}_{m=1,\,m\neq n}^N)$. Finally, inequality (4) is concluded by rearranging the terms in (23).

# F  DISCUSSION ON THE MI RELATED TO $\mathcal{L}_{\text{MUSCLE}}^n(\boldsymbol{z}_i^n)$ AND $\mathcal{L}_{\text{PAIRWISE}}^n(\boldsymbol{z}_i^n)$

It is shown by Oord et al. (2019); Tian et al. (2020) that $\mathcal{L}_{\text{InfoNCE}}^{n,m}(\boldsymbol{z}_i^n)$ in (1) is related to the MI $I(\boldsymbol{z}_i^n; \boldsymbol{z}_i^m)$ as follows:

$$I(\boldsymbol{z}_i^n; \boldsymbol{z}_i^m) \geq \log(B) - \mathbb{E}\mathcal{L}_{\text{InfoNCE}}^{n,m}(\boldsymbol{z}_i^n). \tag{24}$$

Using (24), for $\mathcal{L}_{\text{Pairwise}}^n(\boldsymbol{z}_i^n)$, we have:

$$\mathbb{E}\mathcal{L}_{\text{Pairwise}}^n(\boldsymbol{z}_i^n) = \sum_{m\in[N]\backslash\{n\}}\mathbb{E}\mathcal{L}_{\text{InfoNCE}}^{n,m}(\boldsymbol{z}_i^n)$$

$$\geq (N-1)\log(B) - \sum_{m\in[N]\backslash\{n\}}I(\boldsymbol{z}_i^n; \boldsymbol{z}_i^m). \tag{25}$$

By rearranging the terms in (25), we have the following inequality for the pairwise alignment:

$$\sum_{m\in[N]\backslash\{n\}}I(\boldsymbol{z}_i^n; \boldsymbol{z}_i^m) \geq (N-1)\log(B) - \mathbb{E}\mathcal{L}_{\text{Pairwise}}^n(\boldsymbol{z}_i^n). \tag{26}$$

Based on inequality (26), minimizing $\mathcal{L}_{\text{Pairwise}}^n(\boldsymbol{z}_i^n)$ is equivalent to maximizing the MI $\sum_{m\in[N]\backslash\{n\}}I(\boldsymbol{z}_i^n; \boldsymbol{z}_i^m)$. This pairwise MI formulation lacks conditional MI. In other words, the other representation vectors $\boldsymbol{z}_i^{m'}$, where $m' \in [N] \backslash \{m, n\}$, provide no additional information

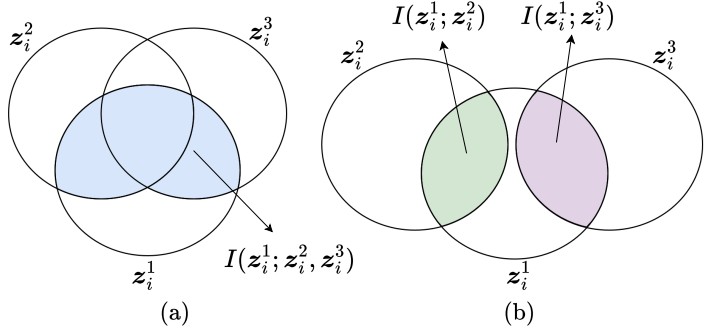

Figure 7: Venn diagrams illustrating the MI $I(z_i^1; z_i^2, z_i^3)$ among three variables $z_i^1$, $z_i^2$, and $z_i^3$. (a) In general, $I(z_i^1; z_i^2, z_i^3) = I(z_i^1; z_i^2) + I(z_i^1; z_i^3 \,|\, z_i^2)$. Since $z_i^1$, $z_i^2$, and $z_i^3$ are not independent of one another, $I(z_i^1; z_i^2, z_i^3)$ does not equal $I(z_i^1; z_i^2) + I(z_i^1; z_i^3)$. (b) A special case where $I(z_i^1; z_i^2, z_i^3) = I(z_i^1; z_i^2) + I(z_i^1; z_i^3)$. However, this case cannot occur because $z_i^1$, $z_i^2$, and $z_i^3$ are positives originating from the same data sample $x_i$. Since our proposed Muscle loss function is related to $I(z_i^1; z_i^2, z_i^3)$, it more effectively captures dependencies between the representation vectors.

about the pair $(z_i^n, z_i^m)$. However, we know that all the positives $z_i^n$, $n \in [N]$, originate from the same data sample $x_i \in \mathcal{D}$, and thus, they should share some joint dependencies. This MI perspective further confirms that pairwise alignment approaches cannot fully capture the dependencies among the representation vectors in multi-model scenarios.

On the other hand, as shown in Theorem 1, minimizing our proposed Muscle loss function is equivalent to maximizing the MI $I(z_i^n; \{z_i^m\}_{m=1, m \neq n}^N)$. Figure 7 illustrates that $I(z_i^n; \{z_i^m\}_{m=1, m \neq n}^N)$ and $\sum_{m \in [N] \setminus \{n\}} I(z_i^n; z_i^m)$ are not generally equal. In particular, by using the chain rule for MI (Thomas & Joy, 2006), we have: $I(z_i^n; \{z_i^m\}_{m=1, m \neq n}^N) = \sum_{m=1}^N I(z_i^n; z_i^m \,|\, \{z_i^{m'}\}_{m'=1, m' \neq n}^{m-1})$. The conditional MI in this formulation captures the dependencies among all the representation vectors $z_i^n$, $n \in [N]$, thereby facilitating more effective knowledge transfer among them.

# G DETAILS OF THE LOCAL TRAINING AND TEST DATASETS IN EXPERIMENTS

We consider that users employ heterogeneous pre-trained FMs tailored to their respective tasks. These pre-trained FMs can be fine-tuned for users' specific tasks, allowing the size of local datasets to remain small. Additionally, we assign different local dataset sizes to users based on the complexity of their tasks. Specifically:

- Each user performing the IC10 task has 50 training samples and 4000 test samples.
- Each user performing the IC100 task has 100 training samples and 2000 test samples.
- Each user performing the MLC task has 200 training samples and 1000 test samples.
- Each user performing the SS task has 1000 training samples and 3000 test samples.
- Each user performing the TC task has 100 training samples and 5000 test samples.

The data samples are uniformly divided across users with the same task.

# H DETAILS OF THE SELECTED MODELS AND ASSIGNED TASKS FOR USERS

In Setup1, we consider six users and assign their tasks and FMs as follows:

- Users 1, 2, and 3 perform the MLC task using ViT-Base, ViT-Small, and ViT-Large, respectively.
- Users 4 and 5 perform the IC100 task using ViT-Base and ViT-Small, respectively.
- User 6 performs the IC10 task using ViT-Tiny.

In Setup2, in addition to the six users from Setup1, we consider four more users with the following tasks and FMs:

- Users 7 and 8 perform the SS task using SegFormer-B0 and SegFormer-B1, respectively.

- Users 9 and 10 perform the TC task using BERT-Base and DistilBERT-Base, respectively.

For users performing the MLC, IC100, and IC10 tasks, we use two linear layers with a rectified linear unit (ReLU) in between as the task-specific prediction head. For users performing the SS task, we use the multi-layer perceptron (MLP) decoder proposed by Xie et al. (2021) as their task-specific prediction head. For users performing the TC task, we use a linear layer as the task-specific prediction head.

## I    IMPACT OF DECREASING MUSCLE LOSS ON $\Delta$

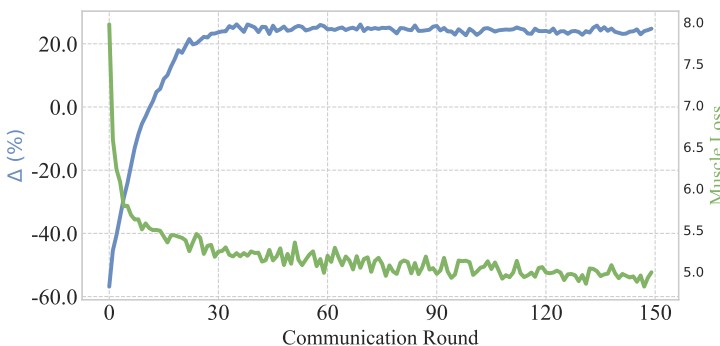

Figure 8: Muscle loss and $\Delta$ evolution over communication rounds. Pascal VOC is used as the public dataset.

## J    PERFORMANCE OF FEDMUSCLE USING A SYNTHETIC PUBLIC DATASET

To generate a synthetic public dataset, we follow existing works such as FedDF (Lin et al., 2020) and employ pre-trained generative models. In particular, we consider that each user generates 50 synthetic images using a pre-trained image-to-image diffusion pipeline based on Stable Diffusion (Rombach et al., 2022), augmented with ControlNet image conditioning (Zhang et al., 2023a) to enable controllable generation aligned with each user's local data. Figure 9 shows one of the synthetic image samples generated by each user in Setup1. The 50 synthetic images generated by each user are then combined to form a shared public dataset consisting of 300 unlabeled synthetic samples. The results in Table 6 demonstrate that FedMuscle can effectively benefit from a synthetic public dataset to align users' representation spaces. Thus, FedMuscle remains effective even when the samples in the shared public dataset are entirely synthetic. These results confirm that synthetic datasets can serve as a viable and practical alternative to real public datasets for FedMuscle.

Table 6: Performance of FedMuscle in Setup1 when the public dataset is obtained by generating 300 synthetic data samples.

| User # | Model | Task | Eval. Metric | FedMuscle (Ours) | Local Training |
|---|---|---|---|---|---|
| 1 | ViT-Base | MLC | micro-F1 | $44.97_{\pm 0.29}$ | $42.17_{\pm 0.24}$ |
| 2 | ViT-Small | MLC | micro-F1 | $49.20_{\pm 0.80}$ | $43.67_{\pm 0.59}$ |
| 3 | ViT-Large | MLC | micro-F1 | $46.90_{\pm 0.16}$ | $42.93_{\pm 0.41}$ |
| 4 | ViT-Base | IC100 | Accuracy | $28.90_{\pm 0.70}$ | $24.77_{\pm 0.42}$ |
| 5 | ViT-Small | IC100 | Accuracy | $30.47_{\pm 0.45}$ | $24.70_{\pm 0.36}$ |
| 6 | ViT-Tiny | IC10 | Accuracy | $61.87_{\pm 0.33}$ | $43.77_{\pm 0.62}$ |
| $\Delta$ (%) ↑ | | | | **+18.32** | 0.00 |

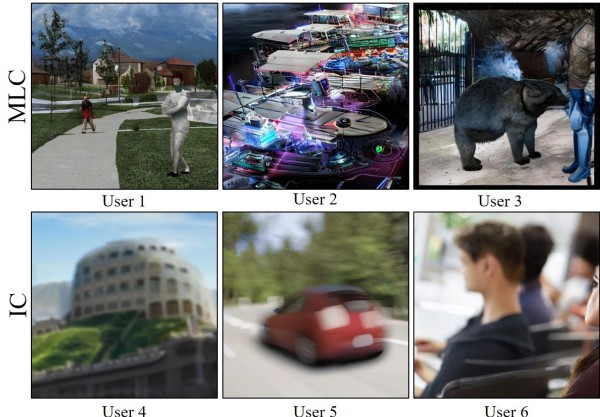

Figure 9: Illustration of a synthetically generated sample for each user in Setup1.

## K   IMPACT OF MUSCLE LOSS FUNCTION IN FEDMUSCLE

Table 7: Performance of FedMuscle in Setup1 using our proposed Muscle loss function compared to the Gramian-based contrastive loss and pairwise alignment.

| Public Dataset | User # | Model | Task | Eval. Metric | FedMuscle | | | Local Training |
|---|---|---|---|---|---|---|---|---|
| | | | | | Muscle Loss (Ours) | Gramian-based Loss | Pairwise Alignment | |
| Pascal VOC | 1 | ViT-Base | MLC | micro-F1 | $46.33_{\pm 0.12}$ | $48.10_{\pm 0.45}$ | $41.90_{\pm 0.08}$ | $42.17_{\pm 0.24}$ |
| | 2 | ViT-Small | MLC | micro-F1 | $49.77_{\pm 0.29}$ | $49.43_{\pm 0.33}$ | $48.07_{\pm 0.33}$ | $43.67_{\pm 0.59}$ |
| | 3 | ViT-Large | MLC | micro-F1 | $49.40_{\pm 0.50}$ | $49.07_{\pm 0.42}$ | $44.57_{\pm 0.49}$ | $42.93_{\pm 0.41}$ |
| | 4 | ViT-Base | IC100 | Accuracy | $36.67_{\pm 0.34}$ | $34.83_{\pm 0.87}$ | $33.53_{\pm 0.17}$ | $24.77_{\pm 0.42}$ |
| | 5 | ViT-Small | IC100 | Accuracy | $29.93_{\pm 0.54}$ | $29.43_{\pm 0.40}$ | $28.20_{\pm 0.36}$ | $24.70_{\pm 0.36}$ |
| | 6 | ViT-Tiny | IC10 | Accuracy | $66.57_{\pm 1.01}$ | $62.47_{\pm 1.80}$ | $61.83_{\pm 0.98}$ | $43.77_{\pm 0.62}$ |
| | $\Delta$ (%) ↑ | | | | **+26.70** | +24.01 | +17.34 | 0.00 |
| COCO | 1 | ViT-Base | MLC | micro-F1 | $49.10_{\pm 0.45}$ | $48.17_{\pm 1.11}$ | $42.27_{\pm 0.33}$ | $42.17_{\pm 0.24}$ |
| | 2 | ViT-Small | MLC | micro-F1 | $51.30_{\pm 0.22}$ | $49.57_{\pm 0.59}$ | $48.53_{\pm 0.54}$ | $43.67_{\pm 0.59}$ |
| | 3 | ViT-Large | MLC | micro-F1 | $50.60_{\pm 0.36}$ | $49.67_{\pm 0.05}$ | $44.67_{\pm 0.12}$ | $42.93_{\pm 0.41}$ |
| | 4 | ViT-Base | IC100 | Accuracy | $37.27_{\pm 0.78}$ | $35.20_{\pm 0.16}$ | $35.13_{\pm 0.31}$ | $24.77_{\pm 0.42}$ |
| | 5 | ViT-Small | IC100 | Accuracy | $30.93_{\pm 0.12}$ | $29.30_{\pm 1.65}$ | $29.20_{\pm 0.57}$ | $24.70_{\pm 0.36}$ |
| | 6 | ViT-Tiny | IC10 | Accuracy | $63.23_{\pm 0.58}$ | $56.77_{\pm 1.23}$ | $61.10_{\pm 1.91}$ | $43.77_{\pm 0.62}$ |
| | $\Delta$ (%) ↑ | | | | **+28.65** | +22.31 | +19.18 | 0.00 |
| CIFAR-100 | 1 | ViT-Base | MLC | micro-F1 | $42.33_{\pm 0.05}$ | $44.13_{\pm 0.79}$ | $41.20_{\pm 0.22}$ | $42.17_{\pm 0.24}$ |
| | 2 | ViT-Small | MLC | micro-F1 | $46.50_{\pm 0.14}$ | $45.80_{\pm 0.73}$ | $46.03_{\pm 0.33}$ | $43.67_{\pm 0.59}$ |
| | 3 | ViT-Large | MLC | micro-F1 | $45.63_{\pm 0.68}$ | $46.53_{\pm 0.33}$ | $43.47_{\pm 0.26}$ | $42.93_{\pm 0.41}$ |
| | 4 | ViT-Base | IC100 | Accuracy | $33.43_{\pm 0.21}$ | $31.70_{\pm 0.54}$ | $31.20_{\pm 0.45}$ | $24.77_{\pm 0.42}$ |
| | 5 | ViT-Small | IC100 | Accuracy | $29.60_{\pm 0.71}$ | $28.47_{\pm 0.66}$ | $25.60_{\pm 1.23}$ | $24.70_{\pm 0.36}$ |
| | 6 | ViT-Tiny | IC10 | Accuracy | $58.37_{\pm 0.57}$ | $56.90_{\pm 2.57}$ | $56.43_{\pm 0.78}$ | $43.77_{\pm 0.62}$ |
| | $\Delta$ (%) ↑ | | | | **+16.88** | +15.19 | +10.48 | 0.00 |

## L   FEATURES OFFERED BY THE MUSCLE LOSS

In this section, we provide a detailed discussion of the features offered by our proposed Muscle loss that are not available in the recently proposed Gramian-based contrastive loss by Cicchetti et al. (2025).

- **Theoretical foundation**: Our proposed Muscle loss is derived from a strong theoretical analysis, whereas the Gramian-based contrastive loss, though effective, is based on intuition and lacks theoretical justification.

- **Consistency with InfoNCE**: For two modalities or models, the Muscle loss reduces to the standard InfoNCE loss, demonstrating that it is a well-structured extension. In contrast, the Gramian-based contrastive loss is not equivalent to the InfoNCE loss in the two-modality (or two-model) case.

- **Well-definedness**: The Gramian-based contrastive loss is defined as follows (Cicchetti et al., 2025): $\mathcal{L}_{\text{Gramian}}^n(\boldsymbol{z}_i^n) = -\log \frac{\exp(-\text{Vol}(\boldsymbol{z}_i^n, \boldsymbol{z}_i^1, ..., \boldsymbol{z}_i^{n-1}, \boldsymbol{z}_i^{n+1}, ..., \boldsymbol{z}_i^N)/\tau)}{\sum_{j=1}^{K} \exp(-\text{Vol}(\boldsymbol{z}_i^n, \boldsymbol{z}_j^1, ..., \boldsymbol{z}_j^{n-1}, \boldsymbol{z}_j^{n+1}, ..., \boldsymbol{z}_j^N)/\tau)}$, where Vol denotes the volume of the $N$-dimensional parallelotope determined by the determinant of the Gramian matrix. However, in the case of $N > d$, the determinant of the Gramian matrix

is zero. Although this condition may not hold in most realistic scenarios, it shows that the Gramian-based contrastive loss is not a well-structured extension for multi-model settings. The Muscle loss does not have such limitations and remains well-defined regardless of the relationship between $N$ and $d$.

- **Insights into temperature parameters**: Our proposed Muscle loss provides new insights into the temperature parameters. Based on the analysis provided in Appendix C, we know, for example, that for three models, the temperature parameters for each pair should be larger than those used for the same pair in a two-model setting.

- **Control on negative transfer**: Some tasks or modalities may be more semantically related to each other than others. Low semantic relevance can lead to negative transfer and degrade performance. The temperature parameters $\tau_{n,m}^{(N)}$, defined between each pair of models in the Muscle loss, provide a flexible and principled mechanism to mitigate negative transfer among tasks with low semantic relevance. This is because semantically closer tasks or modalities can benefit from stronger alignment (lower temperature), while semantically distant ones require softer alignment (higher temperature). In contrast, the Gramian-based contrastive loss uses a single temperature parameter $\tau$ and lacks a mechanism to mitigate negative transfer among tasks or modalities with low relevance.

- **Modularity and integration**: The structure of the Muscle loss enables modular integration into FL settings. As demonstrated in Section 5, when using the Muscle loss, each user transmits only its representation matrices and offloads to the server the part of the loss computation that depends on the representation matrices of other users. The Gramian-based contrastive loss, however, lacks such modularity, resulting in higher computational costs for users.

## M    INTEGRATION OF MUSCLE LOSS INTO CREAMFL

In the CreamFL (Yu et al., 2023) setup, the local models used by the users are as follows:

- ResNet-18 (He et al., 2016) is used by the uni-modal image users.
- GRU (Chung et al., 2014) is used by the uni-modal text users.
- ResNet-18 and GRU serve as the image and text encoders, respectively, for the multi-modal users.

The global model at the server is designed to be larger than those used by the users. In particular, the server employs ResNet-101 and BERT as the image and text encoders, respectively. All models are trained from scratch. In CreamFL, only the representations of the public data obtained from the models are communicated between the users and the server. Specifically, the server generates global image and text representations using the global model and sends them to the users.

Each user first trains its local model using its local dataset and then applies a local contrastive regularization method. In this step, each user employs CL with the InfoNCE loss to align the representations of the public data generated by its own model (e.g., local image representations) with the global representations of the other modality (e.g., global text representations). Additionally, each user aims to make the representations obtained from its own model for a given modality (e.g., image) similar to the global representations of the same modality. To this end, each user employs a loss function similar to the one proposed in MOON (Li et al., 2021a). Then, each user obtains the representations of the public data from its model and sends them to the server.

The server first trains the global model using the public dataset and then applies a global-local contrastive aggregation method. In this step, for each modality, the server assigns a score to the local representations received from the users and aggregates them based on these scores. To compute the score for each local representation, the server calculates the InfoNCE loss for that representation as the anchor, using the global representations of the other modality as positives and negatives. Then, for each modality, the server minimizes the $\ell_2$ distance between the global representations and the aggregated local representations.

The local contrastive regularization and global-local contrastive aggregation methods in CreamFL are heuristic approaches. For example, extending these methods to support more than two modalities

would be challenging. We replace these heuristic approaches with our proposed Muscle loss, which can systematically capture dependencies among representations across any number of modalities. On the user side, we replace local contrastive regularization with minimization of the Muscle loss to align the local representations of each modality (e.g., image) with the global representations of all modalities (e.g., image and text). On the server side, we replace global-local contrastive aggregation with the Muscle loss to align the global representations of each modality with all local representations received from the users. For this experiment, we used the code provided by the CreamFL paper (Yu et al., 2023) and retained all of their original hyperparameters.

## N    EFFECT OF LOCAL EPOCHS, COMMUNICATION ROUNDS, AND CL EPOCHS

Table 8: Performance of FedMuscle under different numbers of local epochs $E$, communication rounds $R$, and CL epochs $T$. The product of local epochs and communication rounds is fixed at 150 (i.e., $E \times R = 150$). The public dataset is derived from COCO.

| User # | Model | Task | Eval. Metric | FedMuscle | | | | | | | | | Local Training |
|---|---|---|---|---|---|---|---|---|---|---|---|---|---|
| | | | | $R=1$ | | | $R=10$ | | | $R=25$ | | | |
| | | | | $T=1$ | $T=3$ | $T=5$ | $T=1$ | $T=3$ | $T=5$ | $T=1$ | $T=3$ | $T=5$ | |
| 1 | ViT-Base | MLC | micro-F1 | $43.73_{\pm 0.45}$ | $44.13_{\pm 0.41}$ | $48.90_{\pm 0.14}$ | $47.33_{\pm 0.21}$ | $48.60_{\pm 0.50}$ | $44.20_{\pm 0.45}$ | $49.03_{\pm 0.33}$ | $49.27_{\pm 0.17}$ | $49.47_{\pm 0.12}$ | $42.17_{\pm 0.24}$ |
| 2 | ViT-Small | MLC | micro-F1 | $45.30_{\pm 0.57}$ | $46.63_{\pm 0.60}$ | $46.77_{\pm 0.69}$ | $50.53_{\pm 0.45}$ | $51.23_{\pm 0.46}$ | $51.07_{\pm 0.19}$ | $51.17_{\pm 0.21}$ | $51.93_{\pm 0.21}$ | $51.23_{\pm 0.26}$ | $43.67_{\pm 0.59}$ |
| 3 | ViT-Large | MLC | micro-F1 | $44.00_{\pm 0.70}$ | $44.23_{\pm 0.68}$ | $45.30_{\pm 0.50}$ | $49.37_{\pm 0.09}$ | $50.33_{\pm 0.34}$ | $50.70_{\pm 0.41}$ | $51.40_{\pm 0.43}$ | $51.57_{\pm 0.12}$ | $51.97_{\pm 0.31}$ | $42.93_{\pm 0.41}$ |
| 4 | ViT-Base | IC100 | Accuracy | $23.83_{\pm 0.17}$ | $25.07_{\pm 0.53}$ | $26.57_{\pm 0.92}$ | $31.87_{\pm 0.48}$ | $34.60_{\pm 0.28}$ | $35.13_{\pm 0.66}$ | $34.40_{\pm 0.16}$ | $36.37_{\pm 0.12}$ | $37.10_{\pm 0.22}$ | $24.77_{\pm 0.42}$ |
| 5 | ViT-Small | IC100 | Accuracy | $26.70_{\pm 0.83}$ | $27.77_{\pm 0.82}$ | $29.50_{\pm 1.02}$ | $31.73_{\pm 0.68}$ | $30.37_{\pm 0.54}$ | $30.17_{\pm 0.83}$ | $31.40_{\pm 0.28}$ | $30.60_{\pm 0.65}$ | $30.27_{\pm 0.90}$ | $24.70_{\pm 0.36}$ |
| 6 | ViT-Tiny | IC10 | Accuracy | $52.53_{\pm 2.12}$ | $58.20_{\pm 1.21}$ | $58.10_{\pm 1.88}$ | $66.60_{\pm 0.93}$ | $65.60_{\pm 0.59}$ | $65.13_{\pm 0.69}$ | $67.53_{\pm 0.05}$ | $65.07_{\pm 0.48}$ | $64.07_{\pm 1.02}$ | $43.77_{\pm 0.62}$ |
| | $\Delta$ (%) $\uparrow$ | | | +5.71 | +10.18 | +12.81 | +25.37 | +27.05 | +27.3 | +28.91 | +29.21 | +29.06 | 0.00 |

## O    NON-IID DATA PARTITION AMONG USERS

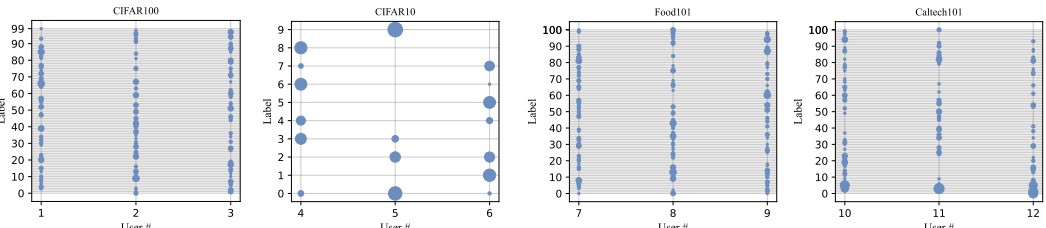

Figure 10: Illustration of the non-IID data distribution among users.

## P    IMPACT OF THE OUTPUT DIMENSION AND TEMPERATURE

We investigate whether tuning the output dimension of the representation models (i.e., $d$) and the temperature parameters (i.e., $\tau_{n,m}^{(4)}$ and $\tau_{n,m}^{(3)}$) as hyperparameters can further improve the performance of FedMuscle. Note that a higher $d$ results in increased communication costs. The results in Figure 11(a) show that the performance remains nearly constant regardless of the chosen output dimension. Additionally, the results in Figure 11(b) show that lower temperature values lead to better overall performance. These findings are consistent with the results presented by Chen et al. (2020) and Yu et al. (2023) using the InfoNCE loss function.

We also investigate the impact of increasing the temperature parameter between tasks with lower semantic relevance in Setup2. The results in Table 9 show that assigning a very large temperature value to user pairs with SS and TC tasks improves the performance of both SS users and TC users. Beyond adjusting the temperature parameters, the semantic relevance between tasks can also be influenced by obtaining representations that enable meaningful knowledge transfer. Exploring other approaches, such as the method proposed in (Mukhoti et al., 2023), may serve as a potential solution for enhancing semantic relevance between SS and TC tasks.

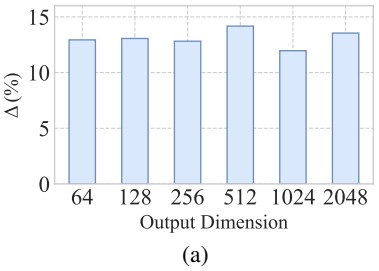 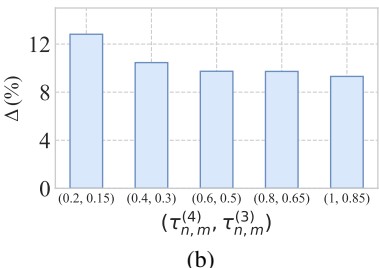

(a)  (b)

Figure 11: Impact of (a) the output dimension of the representation models $d$ and (b) the temperature parameters $\tau_{n,m}^{(4)}$ and $\tau_{n,m}^{(3)}$ on FedMuscle's performance. We set $E = 150$, $R = 1$, and $T = 5$. The public dataset is derived from COCO.

Table 9: Performance of FedMuscle in Setup2 when a large temperature parameter value (i.e., 100) is assigned between users with SS and TC tasks.

| User # | Model | Task | Eval. Metric | FedMuscle (Ours) | Local Training |
|---|---|---|---|---|---|
| | | | | Algorithm | |
| 1 | ViT-Base | MLC | micro-F1 | 47.90 | 42.00 |
| 2 | ViT-Small | MLC | micro-F1 | 50.20 | 43.30 |
| 3 | ViT-Large | MLC | micro-F1 | 49.20 | 42.40 |
| 4 | ViT-Base | IC100 | Accuracy | 31.70 | 24.90 |
| 5 | ViT-Small | IC100 | Accuracy | 27.90 | 24.50 |
| 6 | ViT-Tiny | IC10 | Accuracy | 59.10 | 44.60 |
| 7 | SegFormer-B0 | SS | mIoU | 33.00 | 32.90 |
| 8 | SegFormer-B1 | SS | mIoU | 32.30 | 32.00 |
| 9 | BERT-Base | TC | Accuracy | 44.20 | 40.10 |
| 10 | DistilBERT-Base | TC | Accuracy | 55.80 | 54.90 |
| | $\Delta$ (%) $\uparrow$ | | | **+13.3** | 0.00 |

# Q  DISCUSSION ON POSSIBLE FUTURE WORK DIRECTIONS

**Multi-Modal Representation Learning**  Although most existing self-supervised learning approaches have focused on single-modality inputs (e.g., images), performance can be enhanced by capturing the complementary information available across multiple modalities (Sükei et al., 2024). For example, in (Radford et al., 2021), image captions are considered as an additional modality and CLIP is proposed to learn a multi-modal representation space by jointly training a text and an image encoder. For two modalities, typical CL loss functions, such as InfoNCE, can be used. Multi-modal representation learning using image, text, and 3D point cloud modalities has recently been studied in (Xue et al., 2024a). However, they used the pairwise alignment approach, which cannot fully capture dependencies among the modalities. For more than two modalities, a systematic approach to learning a multi-modal representation space that effectively captures dependencies among modalities is still needed. Recently, Cicchetti et al. (2025) proposed a Gramian-based contrastive loss. Although their approach is effective and intuitive, it lacks theoretical justification and provides no mechanism to address negative transfer among modalities with low semantic relevance. While our primary motivation for deriving the Muscle loss function in (3) was to learn a shared representation space across users' models in FMTL, we believe that the Muscle loss can also enhance performance in multi-modal representation learning. In particular, we can define positives and negatives based on the available modalities and then effectively capture the dependencies among them using our proposed Muscle loss function.

**Performance Metric for Federated Multi-Task Learning**  Users in an FMTL setting have different tasks with distinct performance metrics. Obtaining an overall performance metric across users is challenging due to task heterogeneity. In our experiments, we followed existing works (Lu et al., 2024; Maninis et al., 2019) and used the average per-user performance improvement relative to the local training baseline (i.e., $\Delta$) as the overall performance metric. However, we believe that this metric cannot fully capture the overall performance of an algorithm. This is because a specific algorithm may enhance the performance of only one user and degrade the performance of all other users, yet $\Delta$ may still be positive. For this reason, we presented detailed performance metrics for all users in our experiments. Developing a more informative overall performance evaluation metric for federated multi-task learning could be a valuable future research direction.

**Selection of Representation Matrices in FedMuscle**  In Algorithm 1, we randomly select $M$ representation matrices from other users to compute the aggregated matrix and the weighting coefficient vector for each user. Replacing this random selection with a systematic approach that provides

more relevant knowledge based on each user's task and those of other users would be an interesting direction for future research.

**Convergence Analysis of FedMuscle** In FedMuscle, there is no global model. Thus, new convergence definitions are needed to account for multiple objectives and heterogeneous tasks across users. These definitions may consider the convergence of each user's model individually or a single objective that encompasses all users' models, similar to the Sheaf-FMTL formulation proposed by Issaid et al. (2025). Furthermore, in FedMuscle, each user minimizes two losses: a task-specific loss using its local dataset and a CL loss using a shared public dataset. The convergence of the Muscle loss and its generalization properties should be studied. Overall, our observation is that the CL loss in FedMuscle can be viewed as a regularizer that guides users' local models toward capturing common, task-agnostic information from other users' models, thereby improving their generalization capability. Exploring these theoretical aspects would be an interesting direction for future work. We have provided more insights on FedMuscle's convergence in Appendix S.

# R  LIMITATIONS

- FedMuscle relies on a public dataset. Legal and copyright issues related to the public data should be carefully considered in real-world applications. Similar to typical federated learning algorithms, where users adhere to the global model architecture and training guidelines imposed by the server, in FedMuscle, users should also adhere to the public dataset provided by the server.

- FedMuscle utilizes the Muscle loss on the representations from users' models. Since users have heterogeneous models and diverse tasks, it is important to identify which part of each model should serve as the representation model and which part should function as the task-specific prediction head. This distinction allows users to generate representations that are meaningful and suitable for CL, facilitating the alignment of representations across different users' models.

- We considered multiple CV and NLP tasks in our experimental setup and provided a comparison with multi-modal federated learning algorithms based on their experimental settings. However, we believe that the experimental setup can be further expanded by incorporating more challenging tasks. We will continue to explore this direction in future work.

# S  DISCUSSION ON CONVERGENCE OF FEDMUSCLE

In this section, we show that the Muscle loss in FedMuscle can be considered a regularizer. We then show that, with some simplifications, and for the special case of a linear representation setting, FedMuscle is optimizing an objective nearly identical to the Sheaf-FMTL algorithm (Issaid et al., 2025). Note that the Sheaf-FMTL formulation provides a unified view of several existing FL and FMTL algorithms with guaranteed convergence bounds.

In FedMuscle, we can formulate the overall optimization problem encompassing all user's models as follows:

$$\min_{\substack{\boldsymbol{\theta}^n = \{\boldsymbol{w}^n, \boldsymbol{\phi}^n\}, \\ n \in [N]}} \sum_{n \in [N]} f^n(\boldsymbol{\theta}^n) + g^n(\boldsymbol{w}^n), \tag{27}$$

where we have $f^n(\boldsymbol{\theta}^n) = \mathbb{E}[\mathcal{L}^n(\boldsymbol{\theta}^n)]$ and the expectation is taken over the local dataset $\mathcal{D}^n$ for each user $n \in [N]$. We also have $g^n(\boldsymbol{w}^n) = \mathbb{E}[\mathcal{L}^n_{\text{Muscle}}(\boldsymbol{z}^n_i)]$, where the expectation is taken over the public dataset $\mathcal{D}$. Note that $\boldsymbol{z}^n_i$ is obtained from the output of user $n$'s representation model. We have $\boldsymbol{z}^n_i = e^n(\boldsymbol{w}^n, \boldsymbol{x}_i)$, where $e^n$ is user $n$'s representation model and $\boldsymbol{x}_i \in \mathcal{D}$. In the following, step-by-step, we simplify the expression $g^n(\boldsymbol{w}^n)$.

Employing a simplified version of the contrastive loss has been used in previous studies to facilitate analysis (Xue et al., 2024b; Wang & Liu, 2021). Based on (3), we have

$$
\begin{aligned}
g^n(\boldsymbol{w}^n) &= -\mathbb{E}\log\frac{\alpha_{(i,\ldots,i)}\exp(\boldsymbol{z}_i^n\cdot\sum_{m\in\mathcal{N}^n}\boldsymbol{z}_i^m/\tau_{n,m}^{(N)})}{\sum_{\boldsymbol{j}\in\mathcal{J}^n}\alpha_{\boldsymbol{j}}\exp(\boldsymbol{z}_i^n\cdot\sum_{m\in\mathcal{N}^n}\boldsymbol{z}_{j_m}^m/\tau_{n,m}^{(N)})} \\
&= -\mathbb{E}\log\alpha_{(i,\ldots,i)} - \mathbb{E}[\boldsymbol{z}_i^n\cdot\sum_{m\in\mathcal{N}^n}\boldsymbol{z}_i^m/\tau_{n,m}^{(N)}] \\
&\quad + \mathbb{E}\log\Big(\sum_{\boldsymbol{j}\in\mathcal{J}^n}\alpha_{\boldsymbol{j}}\exp\big(\boldsymbol{z}_i^n\cdot\sum_{m\in\mathcal{N}^n}\boldsymbol{z}_{j_m}^m/\tau_{n,m}^{(N)}\big)\Big) \\
&\overset{(a)}{\approx} -\mathbb{E}\log\alpha_{(i,\ldots,i)} - \mathbb{E}[\boldsymbol{z}_i^n\cdot\sum_{m\in\mathcal{N}^n}\boldsymbol{z}_i^m/\tau_{n,m}^{(N)}] \\
&\quad + \mathbb{E}\max_{\boldsymbol{j}\in\mathcal{J}^n}\Big(\log\alpha_{\boldsymbol{j}} + \big(\boldsymbol{z}_i^n\cdot\sum_{m\in\mathcal{N}^n}\boldsymbol{z}_{j_m}^m/\tau_{n,m}^{(N)}\big)\Big),
\end{aligned}
\tag{28}
$$

where (a) follows from the log-sum-exp approximation. Let $\boldsymbol{j}^{\max}\in\mathcal{J}^n$ be the index tuple that maximizes the last term in (28). The optimization problem (27) is minimized by each user $n$ with respect to its local model parameters $\boldsymbol{\theta}^n$. Thus, in the provided simplified contrastive loss in (28), $\mathbb{E}\log\alpha_{(i,\ldots,i)}$ and $\mathbb{E}\log\alpha_{\boldsymbol{j}^{\max}}$ are constants and can be ignored when optimizing (27). We have

$$
g^n(\boldsymbol{w}^n) \approx -\mathbb{E}[\boldsymbol{z}_i^n\cdot\sum_{m\in\mathcal{N}^n}(\boldsymbol{z}_i^m - \boldsymbol{z}_{j_m^{\max}}^m)/\tau_{n,m}^{(N)}].
\tag{29}
$$

From (29), we observe that $g^n(\boldsymbol{w}^n)$ serves as a regularizer in the optimization problem (27). We also observe that, by choosing large values for the temperature parameters $\tau_{n,m}^{(N)}$, the impact of this regularization term in (27) decreases. In the extreme case where the temperature parameters are set to $\infty$, (27) reduces to local training. In this case, each user optimizes its model solely using its local dataset without collaborating with other users.

Now, let us focus on a linear representation setting in which $e^n(\boldsymbol{w}^n, \boldsymbol{x}_i) = \boldsymbol{W}^n\boldsymbol{x}_i$ for $\boldsymbol{W}^n\in\mathbb{R}^{d\times d^n}$, where $d^n$ denotes the input dimension of user $n$'s representation model based on its task. Note that focusing on a linear representation setting has been widely adopted in FL (Collins et al., 2021) and CL (Nakada et al., 2021; Xue et al., 2024b) analysis. We have $\boldsymbol{W}^n\boldsymbol{x}_i = (\boldsymbol{x}_i^T\otimes I_d)\boldsymbol{w}^n$, where $\otimes$ denotes the Kronecker product and $\boldsymbol{w}^n\in\mathbb{R}^{d.d^n}$ is the vectorized form of the parameters of user $n$'s representation model. Thus, we can rewrite $g^n(\boldsymbol{w}^n)$ in (29) as follows:

$$
g^n(\boldsymbol{w}^n) \approx -\mathbb{E}[(\boldsymbol{x}_i^T\otimes I_d)\boldsymbol{w}^n\cdot\sum_{m\in\mathcal{N}^n}\frac{1}{\tau_{n,m}^{(N)}}[(\boldsymbol{x}_i - \boldsymbol{x}_{j_m^{\max}})^T\otimes I_d]\boldsymbol{w}^m].
\tag{30}
$$

By defining $\boldsymbol{p}^n = \mathbb{E}[\boldsymbol{x}_i^T\otimes I_d]$ and $\boldsymbol{p}^{n,m} = \mathbb{E}[(\boldsymbol{x}_i - \boldsymbol{x}_{j_m^{\max}})^T\otimes I_d]$, we have

$$
\begin{aligned}
g^n(\boldsymbol{w}^n) &\approx -\boldsymbol{p}^n\boldsymbol{w}^n\cdot\sum_{m\in\mathcal{N}^n}\frac{1}{\tau_{n,m}^{(N)}}\boldsymbol{p}^{n,m}\boldsymbol{w}^m \\
&= -\sum_{m\in\mathcal{N}^n}\frac{1}{\tau_{n,m}^{(N)}}(\boldsymbol{p}^n\boldsymbol{w}^n\cdot\boldsymbol{p}^{n,m}\boldsymbol{w}^m).
\end{aligned}
\tag{31}
$$

By combining (27) and (31), we have

$$
\min_{\substack{\boldsymbol{\theta}^n=\{\boldsymbol{w}^n,\boldsymbol{\phi}^n\}, \\ n\in[N]}}\ \sum_{n\in[N]}f^n(\boldsymbol{\theta}^n) - \sum_{n\in[N]}\sum_{m\in\mathcal{N}^n}\frac{1}{\tau_{n,m}^{(N)}}(\boldsymbol{p}^n\boldsymbol{w}^n\cdot\boldsymbol{p}^{m,n}\boldsymbol{w}^m).
\tag{32}
$$

Now, let us compare the optimization problem (32) with the Sheaf-FMTL optimization problem in (Issaid et al., 2025). The Sheaf-FMTL optimization problem is defined as follows for a decentralized setting in which each user is able to collaborate with its neighboring users in $\mathcal{N}^n$ (Issaid et al., 2025):

$$
\min_{\substack{\boldsymbol{\theta}^n=\{\boldsymbol{w}^n,\boldsymbol{\phi}^n\}, \\ n\in[N]}}\ \sum_{n\in[N]}f^n(\boldsymbol{\theta}^n) + \frac{\lambda}{2}\sum_{n\in[N]}\sum_{m\in\mathcal{N}^n}\|\boldsymbol{P}^{n,m}\boldsymbol{\theta}^n - \boldsymbol{P}^{m,n}\boldsymbol{\theta}^m\|^2,
\tag{33}
$$

where $\boldsymbol{P}^{n,m}$ and $\boldsymbol{P}^{m,n}$ are learnable matrices and the term $\boldsymbol{P}^{n,m}\boldsymbol{\theta}^n - \boldsymbol{P}^{m,n}\boldsymbol{\theta}^m$ captures the discrepancy or dissimilarity between local models $\boldsymbol{\theta}^n$ and $\boldsymbol{\theta}^m$. We have $\|\boldsymbol{P}^{n,m}\boldsymbol{\theta}^n - \boldsymbol{P}^{m,n}\boldsymbol{\theta}^m\|^2 = \|\boldsymbol{P}^{n,m}\boldsymbol{\theta}^n\|^2 + \|\boldsymbol{P}^{m,n}\boldsymbol{\theta}^m\|^2 - 2(\boldsymbol{P}^{n,m}\boldsymbol{\theta}^n \cdot \boldsymbol{P}^{m,n}\boldsymbol{\theta}^m)$. By ignoring the regularization terms $\|\boldsymbol{P}^{n,m}\boldsymbol{\theta}^n\|^2$ and $\|\boldsymbol{P}^{m,n}\boldsymbol{\theta}^m\|^2$, the Sheaf-FMTL optimization problem can be reformulated as follows:

$$\min_{\substack{\boldsymbol{\theta}^n=\{\boldsymbol{w}^n, \boldsymbol{\phi}^n\}, \\ n\in[N]}} \sum_{n\in[N]} f^n(\boldsymbol{\theta}^n) - \lambda \sum_{n\in[N]} \sum_{m\in\mathcal{N}^n} \boldsymbol{P}^{n,m}\boldsymbol{\theta}^n \cdot \boldsymbol{P}^{m,n}\boldsymbol{\theta}^m. \tag{34}$$

By comparing the simplified version of the FedMuscle optimization problem in (32) with the Sheaf-FMTL optimization problem in (34), we observe that

- Due to the similarities between these optimization problems, the convergence of FedMuscle would be comparable to that of the Sheaf-FMTL algorithm under the simplified assumptions considered.

- Due to the use of a public dataset in FedMuscle, $\boldsymbol{p}^n$ and $\boldsymbol{p}^{m,n}$ can be obtained from the public dataset based on the hard negative term in the contrastive loss. Thus, unlike in the Sheaf-FMTL algorithm, where $\boldsymbol{P}^{n,m}$ and $\boldsymbol{P}^{m,n}$ are learnable matrices, $\boldsymbol{p}^n$ and $\boldsymbol{p}^{m,n}$ in FedMuscle are determined by the contrastive loss and the public dataset.

- As in the Sheaf-FMTL algorithm, the term $-\boldsymbol{p}^n\boldsymbol{w}^n \cdot \boldsymbol{p}^{m,n}\boldsymbol{w}^m$ in (32) can be viewed as capturing the discrepancy or dissimilarity between $\boldsymbol{w}^n$ and $\boldsymbol{w}^m$. However, in FedMuscle, this term focuses on the representation models rather than on the full model parameters $\boldsymbol{\theta}^n$ and $\boldsymbol{\theta}^m$.

- The Sheaf-FMTL algorithm uses a single hyperparameter $\lambda$ to control the impact of other users' models on each user's local model training. However, in FedMuscle, the temperature parameters $\tau_{n,m}^{(N)}$ arising from the contrastive loss control this impact between each pair of users and can be determined based on the semantic similarity among users' tasks.

## T  THE USE OF LARGE LANGUAGE MODELS (LLMs)

We used LLMs to edit the paper, including grammar, spelling, and word choice. Thus, LLMs did not play a significant role in the research ideation or writing of this paper.

