# OpenReview forum: "Toward Enhancing Representation Learning in Federated Multi-Task Settings"
_ICLR.cc/2026/Conference — ICLR 2026 Poster_

### Official Review · Reviewer_pgBB · 2025-10-20

**Soundness:** 3
**Presentation:** 3
**Contribution:** 3
**Rating:** 6
**Confidence:** 3

**Summary:**

This paper addresses federated multi-task learning (FMTL) with heterogeneous models and tasks. The authors propose Muscle loss, a novel contrastive learning objective that aligns representations from multiple models simultaneously by capturing dependencies among all models' representations. Building on this, they develop FedMuscle, a practical FMTL algorithm where users transmit representation vectors of public data to a server, which computes aggregated matrices and weighting coefficients. The paper demonstrates that minimizing Muscle loss is equivalent to maximizing a lower bound on mutual information among models' representations. Experiments on image and language tasks show consistent improvements over baselines.

**Strengths:**

- The shift from parameter-sharing to representation-alignment is conceptually appealing and well-motivated. The paper addresses a genuine gap in FMTL: most existing methods assume model congruity, while this work handles arbitrary heterogeneous architectures. The focus on learning a shared representation space rather than shared parameters is a valuable perspective.
- The derivation of the Muscle loss from first principles through probability density ratios (Appendix A) demonstrates rigor. Theorem 1 establishing the MI lower bound is meaningful and provides principled justification for the approach. The analysis comparing Muscle loss to pairwise alignment from an MI perspective (Appendix F) clearly shows why capturing joint dependencies matters.
- FedMuscle is well-engineered for federated settings. The communication cost reduction via random selection of M representation matrices (Algorithm 1) is pragmatic. Users don't reveal local model parameters (only representations on public data) which addresses privacy concerns better than some alternatives. The modularity of computation (server computes aggregations while users perform gradient updates) is efficient.
- The paper evaluates across diverse tasks (image classification, semantic segmentation, text classification) with multiple datasets. The comparison with numerous baselines (FedRCL, SAGE, CoFED, FedDF, FedHeNN) is thorough. Setup2's multi-modal evaluation strengthens claims about generality. Integration into CreamFL demonstrates broader applicability. Ablation studies on E, R, T, M, and d are helpful.

**Weaknesses:**

- A fundamental limitation inadequately addressed is how to decompose each heterogeneous model into representation model $w_n$ and task-specific head $\phi_n$. The paper assumes "the output features of all representation models have the same dimension d" and notes this "requirement does not preclude model heterogeneity" and "can be relaxed by appending a lightweight, learnable projection head." This is vague. For practitioners: What if optimal representation dimensions differ by task?, How do you choose which layers constitute the encoder vs. decoder for arbitrary architectures?, Is the projection head frozen or jointly trained? How is its initialization determined?
This choice fundamentally affects what "shared representation space" means and is never empirically investigated.
- FedMuscle critically relies on a shared public dataset D. The paper notes this dataset "may be uni-modal or multi-modal, depending on the users' tasks" and can be "easily obtained." However, in Setup1, using CIFAR-100 (lower performance) vs. COCO (higher performance) shows ~12% performance swing in $\Delta$. For multi-modal Setup2 with mixed CV/NLP tasks, using Flickr30K captions for both modalities seems contrived. Are image captions truly representative of what text classification models need? Real federated scenarios may not have access to high-quality public data relevant to all tasks. The paper doesn't analyze sensitivity to public data quality/task relevance rigorously.
- The multi-modal evaluation in Setup2 mixes semantically distant tasks. Users 1-6 use vision tasks on CV data, while users 9-10 do text classification. The alignment mechanism forces all to learn representations in the same space using Flickr30K captions as the public dataset. This seems artificial. How does aligning a semantic segmentation model with a text classifier via image captions facilitate knowledge transfer? The small improvements for some users (SS tasks in Table 2) may reflect this semantic distance. The paper claims the approach "handles both model and task heterogeneity," but doesn't sufficiently explore when heterogeneity becomes too extreme for representation alignment to be beneficial.
- Some recent SOTA works on FMTL are missing, for instance:

   - Yipan We, Yuchen Zou, Yapeng Li, Bo Du, “Towards Unified Modeling in Federated Multi-Task Learning via Subspace Decoupling”, arXiv preprint arXiv:2505.24185, 2025.
   - Yuxiang Lu, Suizhi Huang, Yuwen Yang, Shalayiding Sirejiding, Yue Ding , Hongtao Lu, “FEDHCA2 : Towards Hetero-Client Federated Multi-Task Learning”, Proceedings of the IEEE/CVF Conference on Computer Vision and Pattern Recognition. 2024.
   - Chaouki Ben Issaid, Praneeth Vepakomma, and Mehdi Bennis. "Tackling Feature and Sample Heterogeneity in Decentralized Multi-Task Learning: A Sheaf-Theoretic Approach." Transactions on Machine Learning Research, 2025.

**Questions:**

Refer to the Weaknesses section.

---

> ### Author Response · Authors · 2025-11-21
> **Response to Reviewer pgBB - Part 1**
>
> We thank the reviewer for their positive feedback and constructive comments. We appreciate that the reviewer recognized that our work addresses a genuine gap in FMTL and viewed the shift toward representation alignment as an appealing and valuable perspective. We are also grateful that the reviewer found the derivation of the Muscle loss rigorous, Theorem 1 provides principled justification, our FedMuscle algorithm is well-engineered for FL and addresses privacy concerns, and our evaluation thorough.
>
> ---
> **Comment 1:** how to decompose each heterogeneous model into representation model $w_n$ and task-specific head $\phi_n$. What if optimal representation dimensions differ by task?, How do you choose which layers constitute the encoder vs. decoder for arbitrary architectures?, Is the projection head frozen or jointly trained? How is its initialization determined? This choice fundamentally affects what "shared representation space" means and is never empirically investigated.
>
> **Authors' Response:**
>
> * We followed a standard and widely adopted definition used in multi-task learning and representation learning to decompose each heterogeneous model into representation model $w_n$ and task-specific head $\phi_n$: For any model, the task-specific head is the last module that maps a representation to a task output (e.g., classifier, segmentation decoder, prediction head). Everything preceding this component is the representation model. This decomposition is applicable to arbitrary architectures because it only requires identifying the final task-specific module, which is always explicitly defined by the task. The remaining parameters naturally constitute the representation model.
> * It is possible that the optimal representation dimension differs across tasks. As mentioned in Footnote 1, in such cases we can append a lightweight, **learnable** projection head that maps each user’s latent representation to a common dimension. This projection head is trained only using the contrastive loss and is initialized randomly. In other words, the representations used to form positive and negative tuples are taken from the output of this projection head. However, when training the model using the task-specific objective during local epochs or when evaluating the model on its task during inference, the projection head is not used, and the original task-specific model architecture remains unchanged. Thus, we can have a representation model with two heads. The outputs of one head (the projection head) are used for contrastive learning, and the outputs of the other head (the prediction head) are used for task-specific training.
> * We would like to mention that using projection heads to align representation dimensions has been widely utilized in multi-modal learning (e.g., [R1]). Moreover, the advantages of using these projection heads have been shown empirically [R2] and theoretically [R3] in the literature. In particular, using a projection head can improve the representation quality of the preceding layer in contrastive learning.
>
> [R1] X. Chen, et al., "X-Fi: A modality-invariant foundation model for multimodal human sensing," in *Proc. Int. Conf. Learn. Represent. (ICLR)*, Singapore, Apr. 2025.
>
> [R2] T. Chen, et al. "A simple framework for contrastive learning of visual representations," In *Proc. Int. Conf. Mach. Learn. (ICML)*, Jul. 2020.
>
> [R3] Y. Xue, et al., "Investigating the benefits of projection head for representation learning," in *Proc. Int. Conf. Learn. Represent. (ICLR)*, Vienna Austria, May 2024.

---

> ### Author Response · Authors · 2025-11-21
> **Response to Reviewer pgBB - Part 2**
>
> **Comment 2:** FedMuscle critically relies on a shared public dataset D. The paper notes this dataset "may be uni-modal or multi-modal, depending on the users' tasks" and can be "easily obtained." However, in Setup1, using CIFAR-100 (lower performance) vs. COCO (higher performance) shows ~12% performance swing in $\Delta$. For multi-modal Setup2 with mixed CV/NLP tasks, using Flickr30K captions for both modalities seems contrived. Are image captions truly representative of what text classification models need? Real federated scenarios may not have access to high-quality public data relevant to all tasks. The paper doesn't analyze sensitivity to public data quality/task relevance rigorously.
>
> **Authors' Response:**
>
> * Using a shared public dataset in FL settings is widely accepted by the FL community. FedMD (NeurIPS 2019), FedDF (NeurIPS 2020), DeSA (ICML 2024), FCCL (CVPR 2022), CreamFL (ICLR 2023), HAMFL (CVPR 2024), and FedHeNN (ICML 2022) are some examples.
> * We mentioned that the public dataset can be easily obtained because (1) FedMuscle relies on an **unlabeled** public dataset, (2) FedMuscle can work well with a public dataset as small as 500 samples (see Figure 4), and (3) the public dataset can be obtained from publicly available datasets or by generating synthetic samples.
> * In the updated manuscript, we have added a new experiment in Appendix J showing that FedMuscle remains effective even when the shared public dataset is entirely *synthetic*. These results confirm that synthetic datasets can serve as a viable and practical alternative to real public datasets for FedMuscle.
> * We believe that our results in Table 1, Table 2, Figure 4, and the newly added experiment in Appendix J clearly show how sensitive FedMuscle is to the public data quality, its size, and its relevance to users' tasks. We agree with the reviewer that using different public datasets (i.e., Pascal VOC, COCO, CIFAR100, and a synthetic dataset) leads to different performance gains in FedMuscle. Our conclusion, which we have added in Lines 374–396, is that FedMuscle remains effective regardless of the chosen public dataset. However, it can further improve users’ overall performance when the public dataset contains *feature-rich* samples. Notably, across all our experimental setups, FedMuscle consistently provides a positive $\Delta$ and outperforms state-of-the-art baselines.
> * In multi-modal Setup 2 with a mix of CV and NLP tasks, **we did not use Flickr30K captions for both modalities**. We used the Flickr30K images for users with CV tasks and the corresponding Flickr30K image captions for users with NLP tasks. Using image captions to obtain representations from text classification models helps transfer semantic information across tasks by aligning these representations with those obtained from CV models using the corresponding images in the public dataset.

---

> ### Author Response · Authors · 2025-11-21
> **Response to Reviewer pgBB - Part 3**
>
> **Comment 3:** The multi-modal evaluation in Setup2 mixes semantically distant tasks. Users 1-6 use vision tasks on CV data, while users 9-10 do text classification. The alignment mechanism forces all to learn representations in the same space using Flickr30K captions as the public dataset. This seems artificial. How does aligning a semantic segmentation model with a text classifier via image captions facilitate knowledge transfer? The small improvements for some users (SS tasks in Table 2) may reflect this semantic distance. The paper claims the approach "handles both model and task heterogeneity," but doesn't sufficiently explore when heterogeneity becomes too extreme for representation alignment to be beneficial.
>
> **Authors' Response:** We believe there may be some confusion here. We do not use image captions or any text classifier for semantic segmentation models. As stated in lines 330–332: “In Setup2, users with CV tasks [such as semantic segmentation and image classification] obtain representations from the *images* in the public dataset, while users with the TC task obtain representations from the image captions.” Figure 1 also illustrates this multimodal setup: **image captions are used for users with the text classification task**, while images from the public dataset are used for users with CV tasks.
>
> In this work, we have considered diverse model architectures, diverse tasks, and diverse datasets across users. As we stated in Appendix R of the updated manuscript (Appendix P in the previous version), the experimental setup can be further expanded by incorporating more challenging tasks. However, we believe that extreme task heterogeneity across users may lead to negative transfer. We have discussed this issue in Appendix L of the updated manuscript (Appendix J in the previous version). In particular, some tasks are more semantically related to each other than others. Low semantic relevance among tasks can lead to negative transfer and degrade performance. We believe that the temperature parameters in our proposed Muscle loss provide a flexible and principled mechanism to mitigate negative transfer among tasks with low semantic relevance.
>
> As an example, based on experimental results from existing work in contrastive learning, it has been observed that greater task relevance correlates with the need for lower temperature values. For instance, in ImageBind [R4], the authors report that a higher temperature performs better for the (image, Inertial Measurement Unit (IMU)) modality pair, whereas a lower temperature is more effective for the (image, audio) modality pair. This aligns with the idea that semantically closer tasks or modalities benefit from stronger alignment (lower temperature), while semantically more distant ones require softer alignment (higher temperature). We will continue to explore this direction in future work.
>
> [R4] R. Girdhar, et al., "ImageBind: One embedding space to bind them all," in *Proc. IEEE/CVF Conf. Comput. Vis. Pattern Recognit. (CVPR)*, Vancouver, Canada, Jun. 2023.
>
> ---
> **Comment 4:** Some recent SOTA works on FMTL are missing.
>
> **Authors' Response:**
> * FedHCA$^2$ mentioned by the reviewer **was already cited in our paper**. Please refer to Lines 039, 043, 106, 362, and 1431.
> * The FedDEA algorithm, mentioned by the reviewer, provides valuable insights on task-level decoupled aggregation to suppress task-irrelevant disturbances and enable the integration of multiple heterogeneous tasks into a unified model. However, it relies on a global model and assumes model congruity across users.
> * The Sheaf-FMTL algorithm, mentioned by the reviewer, provides interesting discussion on how the FMTL objective can be formulated using sheaf Laplacian regularization. However, its algorithm requires all users to share a scaled version of their local models. Although this formulation is interesting, it may not be practical when dealing with large-scale models in FMTL.
> * We have cited both FedDEA and Sheaf-FMTL in the updated manuscript.
>
> ---
> We thank the reviewer for their positive feedback and constructive comments. We hope that our clarifications have addressed the concerns raised regarding the use of projection heads, sensitivity to public datasets, and the use of image captions in FedMuscle.

---

> ### Comment · Reviewer_pgBB · 2025-11-21
> **Official Comment by Reviewer pgBB**
>
> The authors did provide reasonable explanations for some comments, and I thank them for that. However, I still have the following concerns:
> - The claim that this is "applicable to arbitrary architectures" is too strong. For ViT models, is the "task head" just the final linear layer, or does it include LayerNorm? This ambiguity matters for representation quality.
> - The authors mention that the projection head is initialized randomly. What is the distribution? What architecture for the projection head? How many parameters?
> - Thank you for clarifying the confusion about the multi-modal Setup 2 with a mix of CV and NLP tasks. But the fundamental question remains. Does aligning semantic segmentation representations (pixel-level) with text classification representations (document-level) via loosely related image-caption pairs really facilitate meaningful knowledge transfer?
> -  The authors claim temperature parameters can handle semantic distance but provide NO experimental evidence. Figure 11(b) shows overall temperature effects, not task-pair-specific tuning.
> - The authors refer to ImageBind, but ImageBind aligns modalities describing the SAME content (image, audio, text all describe the same scene). In Setup2, semantic segmentation on COCO and text classification on Yahoo Answers have COMPLETELY DIFFERENT semantic content. This comparison doesn't hold.
> - Table 2 results support my concern. Users 7-8 (SS tasks) show minimal improvement, while User 10 performance actually DECREASES. This suggests the method may indeed struggle with semantically distant tasks, but authors don't acknowledge this.

---

> > ### Author Response · Authors · 2025-11-22
> > **Official Comment by Authors**
> >
> > We would like to thank the reviewer for taking the time to review the updated manuscript and our responses to their comments. Our replies to the remaining concerns are as follows:
> >
> > ---
> > **Q1.** The claim that this is "applicable to arbitrary architectures" is too strong. For ViT models, is the "task head" just the final linear layer, or does it include LayerNorm? This ambiguity matters for representation quality.
> >
> > **Authors' Response:** We agree with the reviewer that following the widely adopted approach that we mentioned to decompose an arbitrary model architecture to an encoder and a task-specific head may not necessarily provide representations that are useful for contrastive alignment. Thus, we were included Lines 1461-1466 in Appendix R of the updated manuscript (Appendix P of the previous version) to clarify the importance of representation quality.
> >
> > For the ViT models, we did not consider the LayerNorm as part of the task head. The task head we used was an MLP consisting of two linear layers with a ReLU in between, as indicated in Appendix H.
> >
> > ---
> > **Q2.** The authors mention that the projection head is initialized randomly. What is the distribution? What architecture for the projection head? How many parameters?
> >
> > **Authors' Response:** The initialization of the projection head can follow PyTorch’s default initialization, which is the Kaiming Uniform (He Uniform) initialization. The projection head can be a simple linear layer. If the output dimension of the encoder is $d_e$, then the number of parameters in the projection head is $d_e \times d + d$ (assuming the bias term is included), where $d$ is the dimension of the representation vectors used for contrastive alignment.
> >
> > ---
> > **Q3.** Does aligning semantic segmentation representations (pixel-level) with text classification representations (document-level) via loosely related image-caption pairs really facilitate meaningful knowledge transfer?
> >
> > **Authors' Response:** We sincerely thank the reviewer for their follow-up, detailed question. The short answer is **yes, if the representation vectors are obtained from the semantic segmentation model such that their alignment with the text representation vectors facilitates the transfer of meaningful knowledge between them**.
> >
> > Good examples that are closely related to Setup2 in our paper are Patch-Aligned Contrastive Learning (PACL) [R5] and GroupViT [R6]. In [R5], it is shown that a weighted sum across all vision patch embeddings, where the weights are derived from patch-level similarities, facilitates capturing patch-level alignment between the image and text modalities. In [R6], the image representations are obtained from GroupViT by averaging the representations of all output segment tokens, while the text representation is taken from the final output token of the text transformer. For contrastive alignment, a projection head is used for each model to produce the final representations used in the contrastive loss.
> >
> > [R5] J. Mukhoti, et al., "Open vocabulary semantic segmentation with patch aligned contrastive
> > learning," in *Proc. IEEE/CVF Conf. Comput. Vis. Pattern Recognit. (CVPR)*, Vancouver, Canada, Jun. 2023.
> >
> > [R6] J. Xu, et al., "GroupViT: Semantic segmentation emerges from text supervision," in *Proc. IEEE/CVF Conf. Comput. Vis. Pattern Recognit. (CVPR)*, New Orleans, LA, Jun. 2022.
> >
> > ---
> > **Q4.** The authors claim temperature parameters can handle semantic distance but provide NO experimental evidence. Figure 11(b) shows overall temperature effects, not task-pair-specific tuning.
> >
> > **Authors' Response:** We are conducting an ablation study on different values of the temperature parameters between semantic segmentation and the other tasks in Setup2. We will provide you with the results and include them in the updated manuscript as soon as they are available. However, using a mathematical analysis based on Equation (5) in the paper, we would like to clarify why the temperature parameters are important, especially when there is semantic irrelevance among some tasks or when we do not have high-quality representations for certain tasks.
> >
> > Consider an extreme case where there is no semantic relevance between the representations produced by the encoder of user $n$ and those of the other users. In this case, by setting very large values for
> > $\tau^{(N)}_{n,m}$ for all $m$ (ideally $\infty$), the contrastive loss for user $n$ is constant, meaning that user $n$ is only relying on its local training.
> >
> > The above analysis indicates: (1) In an ideal situation where the temperature values are selected properly, FedMuscle should not degrade each user’s performance compared to their local training performance. (2) The selection of representation matrices at the server for each user can be done such that, for each user, we only select the representation matrices from other users that have some semantic relevance. This is discussed in Appendix Q of the updated manuscript (Appendix O in the previous version).

---

> > > ### Author Response · Authors · 2025-11-22
> > > **Official Comment by Authors (Cont.)**
> > >
> > > **Q5.** The authors refer to ImageBind, but ImageBind aligns modalities describing the SAME content (image, audio, text all describe the same scene). In Setup2, semantic segmentation on COCO and text classification on Yahoo Answers have COMPLETELY DIFFERENT semantic content. This comparison doesn't hold.
> > >
> > > **Authors' Response:** In Setup2, the **local dataset** for semantic segmentation users is obtained from COCO, and the **local dataset** for text classification users is obtained from Yahoo! Answers. However, contrastive alignment is performed using the **public dataset (i.e., Flickr30K)**. We believe that if we focus on the contrastive alignment using Flickr30K, our comparison with ImageBind is meaningful, since the representations of all users describe the same content, namely the image-caption pairs in Flickr30K.
> > >
> > > ---
> > > **Q6.** Table 2 results support my concern. Users 7-8 (SS tasks) show minimal improvement, while User 10 performance actually DECREASES. This suggests the method may indeed struggle with semantically distant tasks, but authors don't acknowledge this.
> > >
> > > **Authors' Response:** We acknowledge the reviewer’s observation. In Table 2, we did not tune hyperparameters (such as temperature values) and did not change the way we obtain representation vectors compared to other setups (such as Setup1), in order to demonstrate the capability of FedMuscle with fixed hyperparameters and a consistent procedure across different setups. However, as mentioned, tuning the temperature values and obtaining high-quality representation vectors for the tasks can mitigate the performance degradation noted by the reviewer in Table 2.
> > >
> > > ---
> > >
> > > We would like to thank the reviewer again for their follow-up questions. We hope that we have addressed their remaining concerns satisfactorily.

---

> > > > ### Comment · Reviewer_pgBB · 2025-11-25
> > > > **Official Comment by Reviewer pgBB**
> > > >
> > > > I thank the authors for the follow-up response. Most of my concerns are resolved, hence I have increased my confidence score.

---

> > > > > ### Author Response · Authors · 2025-11-29
> > > > > **Official Comment by Authors**
> > > > >
> > > > > We sincerely thank the reviewer for their updated evaluation and for acknowledging that our clarifications addressed most of their concerns.
> > > > >
> > > > > We have included new results in Appendix P of the updated manuscript showing the impact of increasing the temperature parameter between tasks with low semantic relevance (i.e., semantic segmentation and text classification tasks). As shown in Table 9, using a higher temperature value between these tasks leads to improved performance for users performing those tasks. Thank you again for your positive feedback.

---

### Official Review · Reviewer_L4FP · 2025-10-31

**Soundness:** 3
**Presentation:** 2
**Contribution:** 2
**Rating:** 4
**Confidence:** 4

**Summary:**

This study offers FedMuscle, a new federated multi-task learning (FMTL) approach that deals with model and task heterogeneity by aligning latent representations rather than exchanging model parameters.  The key concept is Muscle loss, a theoretically justified contrastive learning objective that aims to jointly align representations from several diverse clients rather than doing pairwise alignments like standard InfoNCE-based approaches.

**Strengths:**

The paper covers a wide range of model structures and task types.

The MI lower-bound perspective is nicely integrated with the contrastive framework.

Covers both unimodal and multimodal configurations; the results are consistent.

Clear improvement over strong baselines (FedHeNN, CoFED, CreamFL, and Muscle Loss).

**Weaknesses:**

The Muscle loss is simply a weighted multi-view InfoNCE variation; comparable theories exist, such as Gramian losses and multi-view MI maximization.

Dependence on a shared public dataset undermines the privacy argument and limits usefulness in confined contexts.

There are no theoretical assurances for FedMuscle's convergence or stability under customer heterogeneity.

Weak ablations include no sensitivity analysis on critical hyperparameters (τ, M, B) or comparison to adaptive optimizers (FedAdam, FedNova).

Lack of interpretability of "representation alignment" and it is unclear what semantic information is transmitted between tasks or how this affects per-task specialization.

**Questions:**

How sensitive is FedMuscle's performance to the number of clients (N) and sample parameter (M)?

Can muscle loss deal with missing modalities or imbalanced representation spaces across tasks?

How plausible is the premise of a common unlabeled dataset in medical or financial FL contexts?

How does the approach perform under highly skewed non-IID distributions or in asynchronous settings?

Could the authors compare the runtime (in hours or GPU-days) to FedHeNN or FedRCL?

Is there a theoretical bound on the communication cost-performance trade-off?

Is there a demonstrable association between the MI limit and Δ performance in experiments?

---

> ### Author Response · Authors · 2025-11-21
> **Response to Reviewer L4FP - Part 1**
>
> We thank the reviewer for their constructive comments. We are pleased that the reviewer found the MI perspective in our paper to be well integrated with our proposed and theoretically justified contrastive loss, and that our experiments cover a wide range of settings with results showing consistent improvements of our approach over the baselines.
>
> ---
>
> **Comment 1:** The Muscle loss is simply a weighted multi-view InfoNCE variation; comparable theories exist, such as Gramian losses and multi-view MI maximization.
>
> **Authors' Response:** Our discussion in Section I (lines 052–086) clarifies the limitations of existing contrastive losses and explains how our proposed Muscle loss addresses these limitations through **theoretically grounded weighting coefficients**. We have also provided detailed distinctions between the Muscle loss and the recently proposed Gramian-based contrastive loss (ICLR 2025) in Appendix L of the updated manuscript (Appendix J in the previous version). In particular, the following features are offered by our proposed Muscle loss compared with the recently proposed Gramian-based contrastive loss from ICLR 2025:
> * **Theoretical foundation**
> * **Consistency with InfoNCE in two-modality scenarios**
> * **Well-definedness regardless of the relationship between the number of users and the representation dimension**
> * **New insights on temperature parameters**
> * **Control of negative transfer**
> * **Modularity and efficiency in FL**
>
> Additionally, Section 4 and Appendices A–F primarily focus on how we derived the Muscle loss and the novel and important insights it provides compared with existing contrastive losses. Moreover, the results in Figure 2 show that the Muscle loss yields better performance than both pairwise alignment and the Gramian-based loss from ICLR 2025, because it **captures dependencies among representations more effectively and is supported by theoretical analysis consistent with the MI perspective**.
>
> ---
> **Comment 2:** Dependence on a shared public dataset undermines the privacy argument and limits usefulness in confined contexts.
>
> **Authors' Response:** Sharing local model parameters in FL is known to leak substantial information about users' private data [R1], and prior work has shown that pixel- or token-level private data can even be reconstructed through gradient or model inversion attacks [R2, R3]. In contrast, transmitting private knowledge by sending features extracted from a shared public dataset treats local models as black boxes and does not reveal model parameters or architectures. This mechanism provides an additional layer of privacy protection while still enabling effective knowledge transfer. For these reasons, we believe that relying on a public dataset strengthens—rather than undermines—the privacy guarantees in our setting.
>
> [R1] M. Nasr, et al., "Comprehensive privacy analysis of deep learning: Passive and active white-box inference attacks against centralized and federated learning," in *IEEE symposium on security and privacy*, San Francisco, CA, May 2019.
>
> [R2] J. Geiping, et al., "Inverting gradients - how easy is it to break privacy in federated learning?," in *Proc. Advances in Neural Inf. Process. Syst. (NeurIPS)*, Dec. 2020.
>
> [R3] H. Yin, et al., "See through gradients: Image batch recovery via GradInversion," in *Proc. IEEE/CVF Conf. Comput. Vis. Pattern Recognit. (CVPR)*, Vancouver, Canada, Jun. 2021.
>
> ---
> **Comment 3:** There are no theoretical assurances for FedMuscle's convergence or stability under customer heterogeneity.
>
> **Authors' Response:** Our work focuses primarily on the design of a new multi-task/multi-modal contrastive objective and the corresponding theoretical analysis for deriving it, as well as the design and empirical validation of our proposed FedMuscle algorithm. In FedMuscle, there is no global model, and therefore the convergence analysis used in typical federated learning algorithms that rely on a global model or a shared encoder is not applicable. In particular, FedMuscle is user-centric rather than global-centric. Our observation is that the contrastive loss in FedMuscle can be viewed as a regularizer that guides users’ local models toward capturing common, task-agnostic information from other users’ models, thereby improving their generalization capability. We will explore the theoretical aspects of FedMuscle’s convergence in future work and have included this direction in Appendix Q. Furthermore, as mentioned by the reviewer, our experiments cover a "wide range of model structures and task types", as well as "both unimodal and multimodal configurations". We believe that these thorough experiments with various tasks and model architectures empirically demonstrate the stability of FedMuscle under heterogeneous settings.

---

> ### Author Response · Authors · 2025-11-21
> **Response to Reviewer L4FP - Part 2**
>
> **Comment 4:** Weak ablations include no sensitivity analysis on critical hyperparameters (τ, M, B) or comparison to adaptive optimizers (FedAdam, FedNova).
>
> **Authors' Response:** We would like to mention that the **ablation on hyperparameter $M$ was already provided in Appendix M of the previous version** and was placed there due to page limits. In the updated manuscript, we have moved it to the main body of the paper (Section 6.3).
>
> Following the reviewer’s suggestion, we have also included the ablations on hyperparameters $\tau$ and $B$ in Appendix P and Section 6.3, respectively, in the updated manuscript. We have also provided the results in the following tables:
>
>
> **Table T1: Impact of temperature parameters on FedMuscle's performance. We set $E=150$, $R=1$, and $T=5$. COCO is used as the public dataset.**
> | ($\tau^{(4)}_{n,m}$, $\tau^{(3)}_{n,m}$) |(0.2, 0.15)|(0.4, 0.3)|(0.6, 0.5)|(0.8, 0.65)|(1, 0.85)|
> |-|-|-|-|-|-|
> |$\Delta$\%|+12.81|+10.45|+9.73|+9.72|+9.31|
>
>
> **Table T2: Impact of batch size on FedMuscle's performance. We set $E=1$, $R=150$, and $T=1$. COCO is used as the public dataset.**
> |$B$|8|16|32|64|128|
> |-|-|-|-|-|-|
> |$\Delta$\%|+25.36|+28.10|+28.65|+31.56|+31.63|
>
>
> Since FedAdam and FedNova rely on a global model and on averaging local models at the server, a fair comparison with such algorithms is impossible under the heterogeneity settings addressed in our paper, because we consider a challenging setup in which users can have heterogeneous models and tasks.
>
> ---
> **Comment 5:** Lack of interpretability of "representation alignment" and it is unclear what semantic information is transmitted between tasks or how this affects per-task specialization.
>
> **Authors' Response:** Our discussion in lines 151–156 aims to clarify that representation alignment enables users in a federated multi-task learning setting to capture common, task-agnostic information and improve the generalization capability of their models. As a result, each user can achieve higher performance on its task compared to training solely on its local dataset.
>
> One of our motivations for considering representation alignment comes from an example in ImageBind [R4], where an image of a beach can evoke the sound of waves, the texture of sand, or even inspire a poem. By aligning visual, text, audio, and other modality-specific features in a shared representation space, semantic information can be transferred across tasks.
>
> In FedMuscle, features corresponding to similar semantic concepts cluster together in the shared space, demonstrating the interpretability of the alignment. This alignment does not compromise per-task specialization, because task-specific features remain distinguishable due to local training at each user. Empirically, we observe improved performance for each user compared to training solely on local data. This indicates that the shared representation effectively transfers meaningful semantic information while allowing models to retain individual task expertise.
>
>
> [R4] R. Girdhar, et al., "ImageBind: One embedding space to bind them all," in *Proc. IEEE/CVF Conf. Comput. Vis. Pattern Recognit. (CVPR)*, Vancouver, Canada, Jun. 2023.
>
> ---
> **Q1:** How sensitive is FedMuscle's performance to the number of clients (N) and sample parameter (M)?
>
> **Authors' Response:** We had already included the ablation on $M$ in Appendix M of the previous manuscript. In the revised manuscript, we have moved the results of this ablation study to the main body of the paper (Section 6.3).
>
> In addition to the results previously provided in Table 1 (for 6 users), Table 2 (for 10 users), and Table 3 (for 35 users), we have conducted a new experiment with non-IID local data distributions across users and provided results for different numbers of users in this setup to show that FedMuscle provides consistent improvement regardless of the number of users $N$. The detailed results can be found in Table 4 of the updated manuscript, and are summarized in the following table:
>
> **Table T3: Impact of number of users on FedMuscle's performance. We set $E=1$, $R=150$, and $T=1$. Pascal VOC is used as the public dataset.**
> |$N$|4|8|12|
> |-|-|-|-|
> |$\Delta$\%|+15.14|+10.18|+17.40|

---

> ### Author Response · Authors · 2025-11-21
> **Response to Reviewer L4FP - Part 3**
>
> **Q2:** Can muscle loss deal with missing modalities or imbalanced representation spaces across tasks?
>
> **Authors' Response:** Yes, our proposed Muscle loss can deal with missing modalities and imblanaced representation spaces across tasks.
>
> The local datasets across users may correspond to different modalities; that is, some users do not have access to all modalities. In FedMuscle, each user trains its own model, and the Muscle loss aligns all users into a shared representation space using a public dataset, not the users' private modalities. This means that the alignment step does not depend on all users having the same modalities, and therefore missing modalities do not break the system. The empirical results in Table 2 and Table 3 further confirm that FedMuscle remains effective even when users have heterogeneous or missing modalities. Additionally, if some samples in the public dataset lack certain modalities, the Muscle loss can still operate correctly by forming positive and negative tuples only from the available modalities for those samples. Thus, the contrastive alignment remains well defined even when some modalities are missing.
>
>
>
>
> In general, contrastive learning is an effective approach to address imbalanced representation spaces caused by the modality gap or class imbalance. Thus, our proposed Muscle loss, along with its specific features, can help to mitigate these imbalanced representation spaces. For example, in [R5], it is mentioned that the temperature parameter in the InfoNCE loss plays an important role in mitigating the modality gap in image-text models like CLIP. We also believe that the temperature parameters in our proposed Muscle loss are important factors for mitigating the modality gap and controlling negative transfer among tasks. This matter was discussed in Appendix J of the previous version (Appendix L in the updated manuscript). Additionally, contrastive learning can also be employed to tackle class imbalance arising in applications such as medical images. Incorporating class labels in the contrastive loss, as has been utilized in [R6, R7], is also applicable to our proposed Muscle loss.
>
> [R5] C. Yaras, et al., "Explaining and mitigating the modality gap in contrastive multimodal learning," in *Conf. on Parsimony and Learn. (CPAL)*, Stanford, CA, Mar. 2025.
>
> [R6] Y. Marrakchi, et al., "Fighting class imbalance with contrastive learning," in *Int. Conf. on Med. Image Comput. Computer-Assist. Interv. (MICCAI)*, Strasbourg, France, Sept. 2021.
>
> [R7] D. Mildenberger, et al. "A Tale of two classes: Adapting supervised contrastive learning to binary imbalanced datasets." In *Proc. IEEE/CVF Conf. Comput. Vis. Pattern Recognit. (CVPR)*, Nashville, TN, Jun. 2025.
>
>
>
> ---
> **Q3:** How plausible is the premise of a common unlabeled dataset in medical or financial FL contexts?
>
> **Authors' Response:** This premise is plausible and already utilized in the literature. For example, FLamby [R8] is a publicly available dataset encompassing seven healthcare datasets. Moreover, AIMHI was introduced in [R9] as a federated co-training technique that uses a public unlabeled dataset and is suitable for healthcare. In particular, [R9] mentions that large public health databases are quite common, citing examples like the US NCHS databases, the UK's NHS databases, the UK Biobank, the MIMIC-III database, and the planned European EHDS. Additionally, public datasets are also available for federated learning in finance. The Elliptic dataset, the Lending Club dataset, and the Kaggle Credit Card Fraud dataset are some examples of publicly available datasets in finance [R10].
>
> Finally, we would like to emphasize that the public dataset used in FedMuscle is unlabeled and its size can be as low as 500 samples (See Figure 4). These factors make the premise of obtaining such public dataset more realistic and practical for real-world federated learning scenarios in healthcare and finance.
>
> [R8] J. O. d. Terrail, et al., "FLamby: Datasets and benchmarks for cross-silo federated learning in realistic healthcare settings," in *Proc. of Conf. Neural Inf. Process. Syst. (NeurIPS)*, New Orleans, LA, Dec. 2022.
>
> [R9] A. Abourayya, et al., "AIMHI: Protecting sensitive data through federated co-training," in *Workshop on Federated Learning: Recent Advances and New Challenges in Conjunction with NeurIPS*, New Orleans, LA, Dec. 2022.
>
> [R10] R. Kang, et al., "Federated machine learning in finance: A systematic review on technical architecture and financial applications," *Applied and Computational Engineering*, Nov. 2024.

---

> ### Author Response · Authors · 2025-11-21
> **Response to Reviewer L4FP - Part 4**
>
> **Q4:** How does the approach perform under highly skewed non-IID distributions or in asynchronous settings?
>
> **Authors' Response:** Our proposed approach performs well under those settings because:
>
> 1. As noted in Section 1 (Lines 040–041), handling non-IID data across users who perform the same task is considerably more manageable than scenarios in which users undertake different tasks. Furthermore, the sampling parameter $M$ in FedMuscle, where $M$ representation matrices are randomly selected for each user, demonstrates that FedMuscle can operate reliably in asynchronous settings. This naturally covers cases in which the server has access to representation matrices from only a subset of users who have completed their local training.
> 2. The results provided in Table 3 were related to a setting in which a Dirichlet distribution with $\alpha=0.1$ was used for non-IID data partition across users with the same task.
> 3. To further demonstrate the capability of FedMuscle under highly skewed non-IID distributions, we have conducted a new experiment in which users 1, 2, and 3 perform image classification on CIFAR-100, users 4, 5, and 6 perform image classification on CIFAR-10, users 7, 8, and 9 perform image classification on Food 101, and users 10, 11, and 12 perform image classification on Caltech 101. A Dirichlet distribution with $\alpha=0.1$ is used for non-IID data partition across users with the same task. More details about this experiment and its results can be found in Section 6.3 of the updated manuscript. We also provide the summary of results in the following table:
>
> **Table T4: FedMuscle's performance in a non-IID setting. We set $E=1$, $R=150$, and $T=1$. Pascal VOC is used as the public dataset.**
>
> | User \#|1|2|3|4|5|6|7|8|9|10|11|12|$\Delta$\%|
> |-|-|-|-|-|-|-|-|-|-|-|-|-|-|
> |Local Training|45.2|54.2|30.8|94.2|94.8|79.2|40.0|36.2|13.6|78.7|75.8|62.6|0.00
> |FedMuscle (Ours)|57.7|54.2|41.4|95.1|92.8|86.0|47.8|40.3|25.6|81.8|78.0|71.0|**+17.40**|
> ---
> **Q5:** Could the authors compare the runtime (in hours or GPU-days) to FedHeNN or FedRCL?
>
> **Authors' Response:** We reported computation costs using average FLOPs in Table 1 because FLOPs offer a more reliable and hardware-independent measure than runtime, which can vary significantly with execution environments. As per reviewer’s suggestion, we have also included a runtime comparison with FedHeNN and FedRCL in the following table for completeness:
>
> **Table T5: Runtime comparison.**
>
> |Metric\Method|FedHeNN|FedRCL|FedMuscle, $D=100$|FedMuscle, $D=500$|FedMuscle, $D=1000$|FedMuscle, $D=3000$|FedMuscle, $D=5000$|
> |-|-|-|-|-|-|-|-|
> |$\Delta$%|+2.57|+2.17|+15.33|+26.01|+28.57|+29.29|+28.65|
> |Computation Cost (TeraFLOPS)|34|78|40|80|129|326|523|
> |GPU-hours|1.6|3.9|2.0|3.7|6.6|14.9|24.5|
>
> As shown in the table, for public dataset sizes of 100 and 500, the computational cost of FedMuscle is comparable to that of FedHeNN and FedRCL. However, FedMuscle significantly improves performance compared with these algorithms.
>
> ---
> **Q6:** Is there a theoretical bound on the communication cost-performance trade-off?
>
> **Authors' Response:** We had provided the communication cost analysis of FedMuscle in Section 5 (Lines 302-314). Based on the provided analysis, the total communication cost of FedMuscle per communication round is $NT[d+B^{M-1}(d+1)]|\mathcal{D}|$. Thus, parameters such as the number of users $N$, the number of CL epochs $T$, output dimension $d$, batch size $B$, the number of selected representation matrices in the Muscle loss $M$, and the size of the public dataset $|\mathcal{D}|$ affect the communication cost. Among these parameters, there is no theoretical bound on how increasing $N$, $d$, or $|\mathcal{D}|$ would affect FedMuscle's performance. However, based on Theorem 1, we know that increasing $B$, $M$, and $T$ can be effective in providing better performance. In particular, we can say that:
> * By increasing $B$, the approximation used for deriving inequality (23) in Appendix E becomes more accurate, and thus minimizing the Muscle loss provides a tighter lower bound for increasing the mutual information. This is empirically confirmed with the results provided in Figure 5.
> * In general, from information theory, we know that $I(z_1;z_2,\ldots,z_M)\geq I(z_1;z_2,\ldots,z_{M-1})$. Thus, increasing $M$ provides more knowledge to each user and we expect to see better performance. This is empirically confirmed with the results provided in Table 5.
> * The number of CL epochs $T$ can influence how effectively the Muscle loss decreases in each communication round, which corresponds to achieving a higher mutual information value. However, this effect can also be achieved by increasing the number of communication rounds. This is empirically confirmed with the results provided in Figure 3.

---

> ### Author Response · Authors · 2025-11-21
> **Response to Reviewer L4FP - Part 5**
>
> **Q7:** Is there a demonstrable association between the MI limit and Δ performance in experiments?
>
> **Authors' Response:** Based on Theorem 1, minimizing the Muscle loss is equivalent to maximizing a lower bound to the mutual information. Thus, we can infer the association of mutual information and $\Delta$ by seeing the association between the Muscle loss and $\Delta$. To answer this question of the reviewer, we have provided a training curve in Appendix I of the updated manuscript. This figure shows how the Muscle loss and $\Delta$ are changing over communication rounds. This figure clearly shows that there is a consistent trend between decreasing the Muscle loss and increasing $\Delta$.
>
> ---
> We thank the reviewer for their constructive comments. We hope that our clarifications have addressed the concerns raised regarding the availability of a public dataset in medical or financial FL contexts, the provision of more experiments on hyperparameters $\tau$, $M$, $B$, $N$, and non-IID distributions, and the capabilities of the Muscle loss in handling missing modalities or imbalanced representation spaces across tasks.

---

> ### Comment · Reviewer_L4FP · 2025-11-27
> **Update about my review**
>
> I would like to thank the reviewer for answering all my questions and comments.
>
> I also acknowledge with the previous reviewer (BYRu) that there is no theoretical analysis with Federated learning (FL) or proofs of convergence, which are usually very common in FL-related papers in ICLR. I have seen the response that there is no global model. Still, there are some decentralized federated learning proofs that I'm not sure will be applied here or not.
>
> Can you tell me if the proof of decentralized FL is applicable here?
>
> Without the FL proof, I am still not sure about updating my score, but I am satisfied with the practical results and the conclusions. Therefore, I have reduced my confidence score and increased the contribution score

---

> > ### Author Response · Authors · 2025-11-29
> > **Official Comment by Authors**
> >
> > We would like to thank the reviewer for taking the time to review the updated manuscript and our responses to their comments. We are grateful that the reviewer is satisfied with our answers to all their questions and comments.
> >
> > **To address the reviewer's remaining concern regarding the convergence of FedMuscle, we have included Appendix S in the updated manuscript.** In this appendix, we show that, under certain simplifying assumptions, FedMuscle's objective is closely related to the Sheaf-FMTL formulation presented in [R11]. The Sheaf-FMTL algorithm is a decentralized FL method that provides a unified view of several existing FL and FMTL algorithms with guaranteed convergence bounds. Appendix S also offers novel insights into why the Muscle loss can be viewed as a regularizer in FedMuscle and how incorporating it into the objective function captures the dissimilarity between users' representation models.
> >
> > Compared with recent CL-based FL algorithms, such as CreamFL (ICLR 2023), which do not provide any convergence proof, we believe that Appendix S offers a good starting point for investigating the convergence of such algorithms in general. We would like to thank the reviewer again for their follow-up question, as well as for their positive feedback and updated evaluation.
> >
> >
> > [R11] C. B. Issaid, et al., "Tackling feature and sample heterogeneity in decentralized multi-task learning: A sheaf-theoretic approach," in *arXiv preprint arXiv:2502.01145*, Jun. 2025.

---

### Official Review · Reviewer_BYRu · 2025-11-01

**Soundness:** 3
**Presentation:** 3
**Contribution:** 3
**Rating:** 6
**Confidence:** 3

**Summary:**

The authors propose an interesting idea of learning a shared representation space in federated multi-task learning settings. The proposed Muscle loss effectively captures dependencies across tasks, and the FedMuscle framework outperforms state-of-the-art baselines, demonstrating the effectiveness of the proposed method.

**Strengths:**

1. The authors provide a solid theoretical analysis of the proposed Muscle loss function.

2. The proposed FedMuscle algorithm can be applied to various tasks, including computer vision and natural language processing, demonstrating its capability to handle model and task heterogeneity in federated learning.

3. The contrastive Muscle loss can be seamlessly integrated into multimodal approaches.

**Weaknesses:**

From the perspective of federated learning theory analysis, the current work lacks proofs of convergence and generalization. Of course, this would be a substantial undertaking, and perhaps more in-depth theoretical analysis in this regard can be considered and refined in future research.

**Questions:**

Please see weaknesses.

**Details Of Ethics Concerns:**

N/A.

---

> ### Author Response · Authors · 2025-11-21
> **Response to Reviewer BYRu**
>
> We thank the reviewer for their positive feedback and constructive comments. We appreciate that the reviewer found our idea of learning a shared representation space in federated learning interesting, our theoretical analysis solid, and that our proposed Muscle loss can be incorporated to a wide range of multi-task and multi-modal approaches.
>
> ---
> **Comment 1:** From the perspective of federated learning theory analysis, the current work lacks proofs of convergence and generalization. Of course, this would be a substantial undertaking, and perhaps more in-depth theoretical analysis in this regard can be considered and refined in future research.
>
> **Authors' Response:** We thank the reviewer for highlighting the importance of theoretical analysis for federated learning convergence. Our work focuses primarily on the design of a new multi-task/multi-modal contrastive objective and the corresponding theoretical analysis for deriving it, as well as the design and empirical validation of our proposed FedMuscle algorithm. As pointed out by the reviewer, providing a rigorous convergence analysis for our proposed federated multi-task learning approach is a substantial undertaking. This is due to the following reasons:
>
> 1.  **No global model**: Typical federated learning algorithms that assume model congruity focus on the convergence of a single objective. In other words, they study the rate at which the system converges to a stable global model or a shared encoder given the users’ updated local models [R1, R2]. However, in our problem, there is no global model. This makes the convergence analysis more complex, as our setting is user-centric rather than global-centric. Therefore, new convergence definitions that account for multiple objectives and heterogeneous tasks across users must be considered.
>
> 2.  **Open problems in contrastive convergence**: In our proposed approach, during each communication round, each user minimizes two losses: a task-specific loss using its local dataset and a contrastive loss using the shared public dataset. The convergence rate of the contrastive loss has been studied in simple scenarios such as multi-view settings, which involve only a single model [R3]. However, in more complex scenarios such as multi-task or multi-modal settings, where there is one model corresponding to each task or modality, the convergence and generalization properties of the contrastive loss remain open problems, particularly for settings with more than two modalities or non-linear representation models [R4, R5, R6].
>
>
> Our observation is that the contrastive loss in our federated multi-task learning approach can be viewed as a regularizer that guides users’ local models toward capturing common, task-agnostic information from other users’ models, thereby improving the generalization capability of their models. We will explore these theoretical aspects in future work and have included them among the possible future directions in Appendix Q of the updated manuscript.
>
>
>
> [R1] X. Li, et al., "On the convergence of FedAvg on non-IID data," In *Proc. Int. Conf. Learn. Represent. (ICLR)*, Apr. 2020.
>
> [R2] M. Setayesh, et al., "PerFedMask: Personalized federated learning with optimized masking vectors," In *Proc. Int. Conf. Learn. Represent. (ICLR)*, Kigali, Rwanda, May 2023.
>
> [R3] Z. Yuan, et al., "Provable stochastic optimization for global contrastive learning: Small batch does not harm performance," In *Proc. Int. Conf. Mach. Learn. (ICML)*, Baltimore, MD, Jul. 2022.
>
> [R4] R. Nakada, et al., "Understanding multimodal contrastive learning and incorporating unpaired data," In *Proc. Int. Conf. on Artif. Intell. and Stat. (AISTATS)*, Valencia, Spain, Apr. 2023.
>
> [R5] Y. Xue, et al., "Understanding the robustness of multi-modal contrastive learning to distribution shift," In *Int. Conf. Learn. Represent. (ICLR)*, Vienna, Austria, May 2024.
>
> [R6] L. Lin, et al., "A statistical theory of contrastive learning via approximate sufficient statistics," In *Proc. Advances in Neural Inf. Process. Syst. (NeurIPS)*, San Diego, CA, Dec. 2025.
>
> ---
> We thank the reviewer for their positive feedback and constructive comments. We hope that our clarifications have addressed the concern raised regarding the federated learning theoretical analysis of our proposed algorithm.

---

> > ### Author Response · Authors · 2025-11-29
> > **Official Comment by Authors**
> >
> > Dear Reviewer BYRu,
> >
> > We have included Appendix S in the updated manuscript, which provides a detailed mathematical analysis of the convergence of FedMuscle. We hope that the discussions in Appendix S offer sufficient insight into FedMuscle's convergence and address your question.
> >
> > Thank you again for your positive feedback.

---

### Official Review · Reviewer_VhyE · 2025-11-12

**Soundness:** 2
**Presentation:** 2
**Contribution:** 2
**Rating:** 4
**Confidence:** 4

**Summary:**

This paper proposes to learn a shared realistic representation space across tasks in Federated multi-task learning (FMTL) rather than shared model parameters, as most existing approaches assume the use of fully or partially homogeneous models across users, which limits their applicability in realistic settings.

**Strengths:**

The paper indicates a common limiting assumption among federated multitask
learning approaches (model congruity) and proposes a new method
(FedMuscle) to overcome this limitation. The paper justifies its method
via theoretical results showing that their approach maximizes the mutual
information among the models. The paper conducts experiments on both
image (ViT, SegFormer) and text (BERT, DistilBERT) domains to justify
the performance of FedMuscle compared to various baseline algorithms.

**Weaknesses:**

The requirement of a shared public dataset is very strong, particularly for
federated learning scenarios. On page 4, lines 163 to 165, this issue is addressed
suggesting using publicly available datasets or synthetic data samples but there are concerns of model collapse with synthetic data (though said concerns mostly focus on the recursively generated data by the model or
model family itself reinforcing its own biases, and said issue is resolved via
adding non-synthetic data which the local users’ private database would be
non-synthetic). The authors did not focus on any results on their method’s performance on synthetic data, so this is unclear.

**Questions:**

See above

---

> ### Author Response · Authors · 2025-11-21
> **Response to Reviewer VhyE - Part 1**
>
> We thank the reviewer for their constructive comments and for recognizing that our paper addresses a common limiting assumption in federated multi-task learning approaches (i.e., model congruity) by learning a shared representation space across tasks.
>
> ---
> **Comment 1:** The requirement of a shared public dataset is very strong, particularly for federated learning scenarios. On page 4, lines 163 to 165, this issue is addressed suggesting using publicly available datasets or synthetic data samples.
>
> **Authors' Response:** The reviewer raised concerns about the requirement for a shared public dataset, but we argue that this practice is long-established across many FL paradigms and widely adopted in the literature. Moreover, FedMuscle is specifically designed to work with unlabeled, distributionally mismatched, small public datasets, while also providing stronger privacy advantages compared with alternative FL approaches as detailed below:
>
> 1. **Previous literature**: We emphasize that using a shared public dataset (either publicly available or synthetically generated) in FL settings is widely accepted by the FL community. This includes KD-based FL algorithms such as FedMD (NeurIPS 2019), FedDF (NeurIPS 2020), FedAD (ICCV 2021), Fed-ET (IJCAI 2022), DeSA (ICML 2024); FSSL algorithms such as CoFED (Knowledge-Based Systems 2023); CL-based FL algorithms such as FCCL (CVPR 2022), CreamFL (ICLR 2023), HAMFL (CVPR 2024); and other model-agnostic approaches such as FedHeNN (ICML 2022).
>
> 2.  **Unlabeled requirement**: Unlike methods such as FedMD (NeurIPS 2019) that rely on labeled public data, FedMuscle only requires an unlabeled public dataset, removing the need for annotation and significantly lowering the barrier for deployment in real-world FL scenarios.
>
> 3. **Distributional flexibility**: Experimental results (Table 1) show that FedMuscle maintains strong performance even when the public dataset (e.g., Pascal VOC) is distributionally different from clients’ private data, demonstrating robustness to cross-domain and cross-distribution mismatches.
>
> 4. **Small sample requirements**: FedMuscle performs well across a wide range of public dataset sizes (as low as 500), confirming that it remains effective even with very limited public data (Figure 4).
>
> 5. **Privacy advantages**: FL knowledge-transfer methods generally fall into three categories: (i) model/gradient transfer, (ii) feature or data transfer from private local data, and (iii) feature or data transfer from public data. Consistent with prior work, we argue that public data feature transfer offers the most favorable privacy–utility trade-off: clients never share model weights, gradients, or features derived from private data. That is, only information computed from public data is exchanged, ensuring full confidentiality of local models and datasets. In this sense, requiring a public dataset is not only realistic but also enhances user privacy compared with alternative FL paradigms.
>
> ---
> **Comment 2:** There are concerns of model collapse with synthetic data (though said concerns mostly focus on the recursively generated data by the model or model family itself reinforcing its own biases, and said issue is resolved via adding non-synthetic data which the local users’ private database would be non-synthetic).
>
> **Authors' Response:** As the reviewer pointed out, we emphasize that each user’s local dataset is non-synthetic and FedMuscle does not involve recursively generating synthetic data. Specifically, in FedMuscle, the public dataset, whether it is a real publicly available dataset or a synthetically generated one, is fixed and reused across all communication rounds to align the users’ representation spaces. Because the public dataset is not regenerated at each round, FedMuscle avoids the recursive synthetic-data feedback loop that may lead to model collapse. Each user can contribute in generating a synthetic public dataset before the FL training using approaches such as distribution matching [R1], training a generator [R2], or using a pretrained generative adversarial network (GAN) [R3].
>
> [R1] B. Zhao, et al., "Dataset condensation with distribution matching," in *Proc. IEEE/CVF Conf. Comput. Vis. Pattern Recognit. (CVPR)*, Vancouver, Canada, Jun. 2023.
>
> [R2] J. Zhang, et al., "DENSE: Data-free one-shot federated learning," in *Proc. Advances in Neural Inf. Process. Syst. (NeurIPS)*, New Orleans, LA, Dec. 2022.
>
> [R3] T. Lin, et al., "Ensemble distillation for robust model fusion in federated learning," In *Proc. Advances in Neural Inf. Process. Syst. (NeurIPS)*, Dec. 2020.

---

> ### Author Response · Authors · 2025-11-21
> **Response to Reviewer VhyE - Part 2**
>
> **Comment 3:** The authors did not focus on any results on their method’s performance on synthetic data, so this is unclear.
>
> **Authors' Response:** To address the reviewer’s concern, we conducted an additional experiment in which the shared public dataset is fully synthetic. Specifically, for each user in Setup1, we generated 50 synthetic images using a pretrained image-to-image diffusion pipeline [R4, R5]. These 50 synthetic images per user were then combined to form a shared public dataset consisting of 300 synthetic samples. The results in the following table demonstrate that FedMuscle can effectively benefit from a synthetic public dataset to align users’ representation spaces. Thus, FedMuscle can still achieve consistent performance improvements even when using synthetic public data. We have added the details of this experiment, along with a few samples of synthetically generated images and its results in Appendix J of the updated manuscript.
>
>
> **Table T1: Performance of FedMuscle in Setup1 using a synthetic public dataset with 300 samples.**
> |User \#|FedMuscle (Ours)|Local Training|
> |-|-|-|
> |1|44.97|42.17|
> |2|49.20|43.67|
> |3|46.90|42.93|
> |4|28.90|24.77|
> |5|30.47|24.70|
> |6|61.87|43.77|
> |$\Delta$\%|**+18.32**|0.00|
>
>
>
>
> [R4] R. Rombach, et al., "High-resolution image synthesis with latent diffusion models," in *Proc. IEEE/CVF Conf. Comput. Vis. Pattern Recognit. (CVPR)*, New Orleans, LA, Jun. 2022.
>
> [R5] L. Zhang, et al., "Adding conditional control to text-to-image diffusion models." in *Proc. IEEE/CVF Int. Conf. Comput. Vis. (ICCV)*. Paris, France, Oct. 2023.
>
> ---
> We thank the reviewer for their constructive comments. We hope that our clarifications have addressed the concerns raised regarding the practicality of using a shared public dataset and the performance of FedMuscle when using a synthetic public dataset.

---

### Author Response · Authors · 2025-11-21
**Thanks for your time and valuable feedback**

We would like to thank the reviewers for their constructive comments which helped us to improve our work. In the revised manuscript, we have used blue color for text that was changed or added based on the reviewers' comments. The responses to the reviewers' comments are given below. We hope that we have addressed the comments in a satisfactory manner.

---

### Author Response · Authors · 2025-12-03
**Summary Comment for the AC Panel (1/2)**

Dear AC Panel,

We would like to thank the reviewers for their constructive comments, which helped us improve our work. We are grateful that our paper received a positive evaluation from the reviewers. In particular, the reviewers highlighted the following **strengths**:

---
* **The authors provide a solid and rigorous theoretical analysis for the derivation of the proposed Muscle loss** (VhyE: Strength 2, BYRu: Strength 1, pgBB: Strength 2).

* **The mutual information lower-bound perspective is nicely integrated with the contrastive framework. It provides a principled justification for the proposed approach and clearly shows why capturing joint dependencies matters** (VhyE: Strength 2, L4FP: Strength 2, pgBB: Strength 2).

* **The proposed contrastive Muscle loss can be seamlessly integrated into multimodal approaches, and the experiments cover both unimodal and multimodal configurations** (BYRu: Strength 3, L4FP: Strength 3, pgBB: Strength 4).

* **This paper addresses a genuine gap in federated multi-task learning, namely the model congruity assumption** (VhyE: Strength 1, pgBB: Strength 1).

* **The experiments cover a wide range of model architectures and tasks across multiple datasets, demonstrating that the FedMuscle algorithm can be applied to various CV and NLP tasks to handle model and task heterogeneity in federated learning** (VhyE: Strength 3, BYRu: Strength 2, L4FP: Strensgth 1, pgBB: Strength 4).

* **The paper compares FedMuscle’s performance with various baseline algorithms, and the results show clear improvements over strong baselines** (VhyE: Strength 3, L4FP: Strength 4, pgBB: Strength 4).

* **The shift from a parameter-sharing perspective to representation alignment is conceptually appealing, well-motivated, and provides a valuable perspective** (pgBB: Strength 1).

* **FedMuscle is well engineered for federated learning settings in terms of communication cost, computation cost, and addressing privacy concerns** (pgBB: Strength 3).

---
During the discussion period, we actively addressed all comments and questions from the reviewers and updated the manuscript accordingly. We would like to bring the AC’s attention to our summarized responses:

* **Questions regarding the public dataset**
    * Concern about the requirement of a shared public dataset in federated learning (**VhyE: Comment 1**)

    **Authors' Response**: Using a shared public dataset in federated learning settings is widely accepted by the FL community. FedMD (NeurIPS 2019), FedDF (NeurIPS 2020), FCCL (CVPR 2022), FedHeNN (ICML 2022), CreamFL (ICLR 2023), DeSA (ICML 2024), and HAMFL (CVPR 2024) are some examples. Furthermore, our proposed algorithm, FedMuscle, relies on an **unlabeled** public dataset and remains effective even with very **limited public data**, e.g., as few as 500 samples (see Figure 4). These properties, along with the fact that the public data can be obtained from publicly available sources or generated synthetically, make the approach practical.

    * Concern about the privacy due to dependence on a shared public data (**L4FP: Comment 2**)

    **Authors' Response**: It has been shown that sharing local model parameters in typical FL algorithms can leak substantial information about users’ private data, and pixel- or token-level private data can even be reconstructed through gradient or model inversion attacks. In contrast, transmitting private knowledge by sending features extracted from a shared public dataset treats local models as black boxes and does not reveal model parameters or architectures. This mechanism provides an additional layer of privacy protection while still enabling effective knowledge transfer. For these reasons, **we believe that relying on a public dataset strengthens—rather than undermines—the privacy guarantees in our setting**. This point was acknowledged by Reviewer pgBB in Strength 3, and Reviewer L4FP also noted that they were satisfied with the conclusions provided in our rebuttal.

    * Analysing sensitivity of FedMuscle to public data quality (**pgBB: Comment 2**)

    **Authors' Response**: Our results in Table 1, Table 2, Figure 4, and Appendix J clearly show the sensitivity of FedMuscle to the public data quality, its size, and its relevance to users' tasks. Although using different public datasets (i.e., Pascal VOC, COCO, CIFAR100, and a synthetic dataset) leads to different performance gains in FedMuscle, it consistently provides a positive Δ and outperforms state-of-the-art baselines across all our experimental setups. Thus, FedMuscle remains effective regardless of the chosen public dataset; however, it can further improve users' overall performance when the public dataset contains feature-rich samples. Reviewer pgBB mentioned that their concerns were resolved by our responses.

---

> ### Author Response · Authors · 2025-12-03
> **Summary Comment for the AC Panel (2/2)**
>
> * **Questions regarding the public dataset (Cont.)**
>
>     * Plausibility of a common unlabeled dataset in medical or financial FL contexts (**L4FP: Q3**)
>
>     **Authors' Response**: This premise is plausible and already utilized in the literature. For example, FLamby is a publicly available dataset encompassing seven healthcare datasets. Moreover, AIMHI was introduced as a federated co-training technique that uses a public unlabeled dataset and is suitable for healthcare. In particular, large public health databases are quite common, such as: the US NCHS databases, the UK's NHS databases, the UK Biobank, the MIMIC-III database, and the planned European EHDS. Additionally, public datasets are also available for federated learning in finance. The Elliptic dataset, the Lending Club dataset, and the Kaggle Credit Card Fraud dataset are some examples of publicly available datasets in finance. Reviewer L4FP noted that they were satisfied with the conclusions provided in our rebuttal.
>
>     * Concern about model collapse when using synthetic public data (**VhyE: Comment 2**)
>
>     **Authors' Response**: This concern is not applicable to our approach as FedMuscle does not involve recursively generating synthetic data.
>
> ---
> * **Questions regarding FedMuscle convergence**
>     * This is the only weakness mentioned by Reviewer BYRu, and they noted that it can be considered in future research. However, Reviewer L4FP also mentioned it in Comment 3 (**BYRu: Comment 1, L4FP: Comment 3**).
>
>     **Authors' Response**: To address this concern, we have included Appendix S in the updated manuscript. In this appendix, we show that, under certain simplifying assumptions widely used in federated learning (FL) and contrastive learning (CL) analysis, FedMuscle’s objective is closely related to the Sheaf-FMTL formulation and its convergence is guaranteed. Appendix S also offers novel insights into why the Muscle loss can be viewed as a regularizer in FedMuscle and how incorporating it into the objective function captures the dissimilarity between users’ representation models. We believe that the analysis provided in Appendix S not only addresses the concern regarding FedMuscle’s convergence, but also serves as a good starting point for analyzing the convergence of CL-based FL algorithms such as CreamFL (ICLR 2023) and FMCSC (NeurIPS 2024), which have not provided any convergence proof.
>
> ---
> * **Additional experiments requested by reviewers**
>     * Performance of FedMuscle on the synthetic public dataset (**VhyE: Comment 3**)
>
>     **Authors' Response**: We have added the details of this experiment and its results in Appendix J of the updated manuscript.
>
>     * Ablation on $M$, $N$, $B$, and $\tau$ (**L4FP: Comment 4 and Q1**)
>
>     **Authors' Response**: Ablation on $M$ was already available in the initial submission. We had also considered different numbers of users (i.e., $N$) in our experiments in the initial submission. We have added more results on ablation on $N$ (Table 4), the results for ablation on $B$ (Figure 5), and $\tau$ (Figure 11(b)) in the updated manuscript. Reviewer L4FP noted that they are satified with the practical results.
>
>     * Performance on highly skewed non-IID distributions (**L4FP: Q4**)
>
>     **Authors' Response**: Table 3 in the initial submission related to an experiment with non-IID data partition across users. We have also included a new experiment (Table 4) in the updated manuscript to show the performance of FedMuscle under a non-IID data distribution setting. Reviewer L4FP noted that they are satisfied with the practical results.
>
>     * Impact of temperature parameter on the performance of users with semantic distant tasks (**pgBB: Comment 3 in initial review and Q4 during reviewers-authors discussion period**)
>
>     **Authors' Response**: In addition to the provided mathematical analysis showing a large temperature value among users with semantically distant tasks can mitigate performance degradation for them, and the explanations in Appendices L and S, we have included Table 9 in the updated manuscript to further clarify this point empirically. Before adding Table 9, Reviewer pgBB noted that most of their concerns were resolved by our responses.
>
> ---
> Finally, we would like to mention that our work can contribute to advancing the field of contrastive learning, federated learning, multi-modal learning, and multi-task learning. As the results in Figure 2 and Table 3 show, our approach can outperform recently proposed methods such as Gramian-based loss (ICLR 2025) and CreamFL (ICLR 2023) while also providing a theoretically grounded approach rather than heuristic-based ones.

---

### Meta-Review · Area_Chair_No8i · 2026-01-08

**Summary:**

This paper proposes to learn a shared representation space across tasks in Federated multi-task learning (FMTL) settings rather than exchanging model parameters, as most existing approaches assume the use of fully or partially homogeneous models across users, which limits their applicability in realistic settings. The proposed Muscle loss captures dependencies across tasks to jointly align representations from several diverse clients rather than doing pairwise alignments like standard InfoNCE-based approaches. The authors also demonstrate that minimizing Muscle loss is equivalent to maximizing a lower bound on mutual information among models' representations. FedMuscle outperforms state-of-the-art baselines, demonstrating the effectiveness of the proposed method.

**Reviewer Concerns:**

The concerns are around lack of novelty, theoretical justification, requirement of a shared public dataset, privay risk, sensitivity to the data, and empirical completeness. The authors have tried to address most concerns and strengthened the paper by providing additional experimental results requested by the reviewers and tried to argue with the reviewers regarding the public dataset. In addition, the authors have included Appendix S in the updated manuscript to address the reviewers' major concern regarding the convergence of FedMuscle. However, some major concerns as follows have not been fully addressed to me:

[Reviewer VhyE] "The requirement of a shared public dataset is very strong". The authors argue that "this practice is long-established across many FL paradigms and widely adopted in the literature"-- This is ture. However, in the FL community a FL framework without sharing a public dataset is the trend, as sharing a public dataset indeed incurs various disadvantages.

[Reviewer L4FP] "The Muscle loss is simply a weighted multi-view InfoNCE variation; comparable theories exist, such as Gramian losses and multi-view MI maximization" (i.e. lack of novelty in main contribution). The authos didn't directly answer whether it is the case but only argues that the proposed loss is different from the recently proposed Gramian-based contrastive loss in some features, avoiding talk about the similarity in their forms.

**Reviewer Scores:**

It is hard to predict Reviewer VhyE's score (rating 4 confidence 4) because his/her major concern is the shared public dataset, which matters.

Reviewer BYRu's score (rating 6 confidence 3) might not be changed as the quality of this paper does not reach next level.

Reviewer L4FP's score (rating 4 confidence 4) might be slightly increased to 6 as the authors have responded to his/her major concen on the theoretical convergence.

by Reviewer pgBB's score (rating 6 confidence 3) is unlikely to be changed as the reviewer has responded to the authors his/her score has been updated after rebuttal and discussion.

---

### Decision · Program_Chairs · 2026-01-26

Accept (Poster)